# IMP: Benchmarking Image Polysemy in Vision-Language Models

## Abstract

Current vision-language models predominantly use contrastive losses to learn from the co-occurrence of image and text. While effective for certain tasks, this approach assumes semantic equivalence between these two modalities. This assumption runs counter to the diverse meanings that a single image can convey, which in turn may compromise visual understanding. To investigate the impact of this assumption, we introduce a novel dataset: **IMP**, designed to challenge and evaluate vision-language models on image polysemy. Our empirical results reveal that current models fall short in recognizing the multiple semantic dimensions of images, underscoring the need for more robust approaches for learning vision-language representations. Code and data will be made available on publication.

## 1 Introduction

Vision-language models (VLM) have made great strides in recent years by leveraging image caption datasets (Radford et al., 2021; Singh et al., 2022; Li et al., 2023; Changpinyo et al., 2021). The use of captions is highly promising, as the language that accompanies images is an incredibly rich source of supervision; it may, for example, describe both the objects and the relations between them (Lin et al., 2015). In this sense, it is clear that captions are a more natural means of describing images than the typical annotation schemes used for large-scale image datasets. Moreover, approaches for learning from image-text pairs have achieved impressive results with a relatively simple contrastive mechanism that pushes matching pairs together and mismatching pairs apart (Radford et al., 2021; Kim et al., 2021; Li et al., 2021). While effective, this mechanism relies on the strong assumption that the caption text is *descriptive* of the image, which may not be the case for naturally occurring images and their accompanying text.

Multimodality is a well-researched principle (Van Leeuwen, 2015), establishing the intricacies of how modalities interact; noting that even when expressing the "same" meaning, modalities may do so differently to due the affordances of each modality. Critical in this observation is the notion of *sameness*: for curated datasets like MSCOCO (Lin et al., 2015) the annotation process was designed such that the captions are descriptive of the image; however, for naturally occurring image-text pairs it cannot be assumed that this is equally the case. Nonetheless, existing approaches assume that co-occurrence equates to semantic sameness and aim to learn from large collections of web-scraped image-text pairs (Lu et al., 2019; Radford et al., 2021; Changpinyo et al., 2021).

Images may convey different meanings, i.e., they may be *polysemic*, and the meaning they have in communication depends on how they are used (Kress & Van Leeuwen, 2006). This use is partially established by the text that accompanies an image, therefore each pairing of an image with multiple captions may convey a different meaning - the captions *anchor* the image to a meaning. Crucially, during this anchoring process, the text does not have to be descriptive of the image, their pairing may be purely associative; by for instance conveying a similar emotion or alluding to a similar abstract concept. Establishing this principle of image polysemy is useful when we consider the prevalence of contrastive learning in vision-and-language (Radford et al., 2021; Li et al., 2021), as these models are optimised by pulling together matching image-text pairs and pushing apart mismatching pairs, thereby inadvertently pulling together *all* captions paired with an image, as well as the captions for neighbouring images (Song & Soleymani, 2019). Existing approaches for this issue treat the lack of consistency between captions as noise (Santurkar et al., 2022), instead we argue that to make full use of the richness of image-text pairings it is necessary to account for image polysemy.

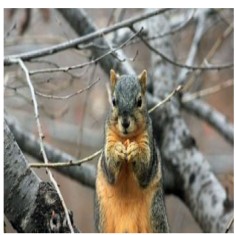 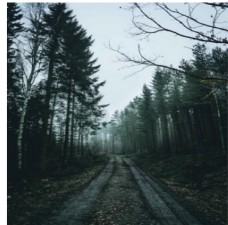 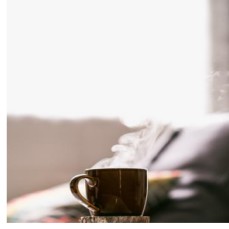 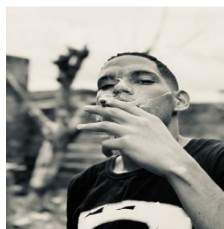

- It's the little thing standing on the branch that count for a biological tribe.
- The secret life of squirrels.
- So tame, he showed no fear, though my lens was feet from his snout.
- A squirrel thinks about its life.
- Hear stories about the mountain's wildlife as you get glazed upon from one of our furry little friends.

- The path in the forest was shrouded in fog.
- The road vanishes into the woods.
- Not surprising me if there was a zombie attack
- I'd rather be walking in a forest than a street.
- Walks lonely deep in the forest.

- Having hot coffee in the morning to begin a new workday.
- Good morning or have a nice day message concept.
- Happy Monday! God will carry you through every storm and give you strength.
- Steam from a morning cup of tea or coffee.
- A woman in her apartment stands by the window, looking at the rain while drinking coffee from a cup.

- A man smoking a cigarette at the site where a populist leader and presidential candidate was assassinated.
- No care in the world.
- Part of the human face smoking a cigarette on the street at night and releasing smoke into the air.
- Man is smoking a cigarette and choking from the smoke.
- A portrait of a brooding man, side view in black.

(a) Examples from IMP

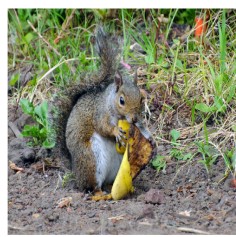 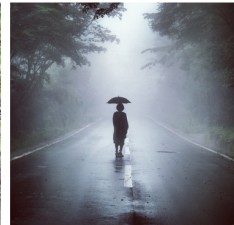 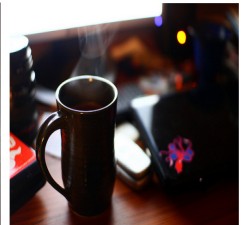 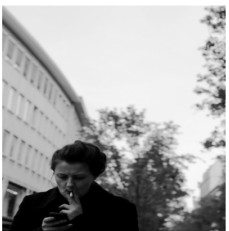

- A squirrel eating a banana peel on the ground.
- A squirrel is eating a banana peel outdoors.
- A brown squirrel eating a rotten banana peel
- A squirrel eating a rotten banana peel in a park.
- A squirrel eating the end an old banana pill.

- A man walking down a rain soaked road holding an umbrella.
- A woman with an umbrella is silhouetted on a foggy road.
- a person with a black umbrella standing in the middle of the road
- A person holds an umbrella in the middle of a rode.
- A person is standing in the middle of a road while its raining.

- a brown cup of steamy coffee on a desk
- A cup of hot coffee on a desk.
- A glorious photograph of steam rising from a mug of coffee.
- A steaming cup of liquid sitting on a desk.
- A hot beverage in a mug on a desk with several other items.

- A full view of an individual in the image
- A person smoking a cigarette next to a tall building.
- THIS IS A BLACK AND WHITE PHOTO OF A WOMAN SMOKING A CIGARETTE LOOKING AT HER PHONE
- A woman is outside smoking a cigarette while looking at her phone.
- A woman smokes a cigarette while looking at her phone.

(b) Visually similar images from MSCOCO

Figure 1: Example images from IMP (top row) and MSCOCO (bottom row). Whilst the datasets contain similar images, the captions for IMP are both descriptive and conceptual, whereas the MSCOCO captions are purely descriptive.

We introduce a novel **IM**age **P**olysemy benchmark, **IMP**, that challenges the prevailing assumption in vision-and-language models. Unlike traditional datasets that focus on descriptive captions, IMP includes diverse captions that range from descriptive to conceptual, thereby embracing the polysemic nature of images. This not only allows for a more nuanced understanding of image-text relationships but also serves as a rigorous benchmark for evaluating the adaptability and robustness of existing models to variations in caption semantics. By doing so, we address a gap in existing vision-and-language research, encouraging the community to move beyond purely descriptive paradigms and explore the rich, multifaceted interplay between visual and textual modalities.

Our contributions are as follows:

- A dataset of images with diverse captions, from descriptive to conceptual, to highlight the polysemic nature of images.
- A large-scale evaluation of existing VLM, exposing the limitations of existing learning paradigms for dealing with image polysemy.

## 2 RELATED WORK

### 2.1 POLYSEMY

The phenomenon of polysemy, where a single form can have multiple meanings can be found for both textual and visual data (Chen et al., 2015; Yao et al., 2018; Saenko & Darrell, 2008). Understanding polysemy is crucial for VLM to achieve robust and nuanced representations across different modalities. Within natural language processing (NLP), polysemy has been studied as Word Sense Disambiguation (WSD) - aimed at resolving the ambiguity of words across contexts (Navigli, 2009). With the rise of large-scale self-supervised pre-training, there have been significant improvements in WSD. In particular, the switch to contextual word embeddings has resulted in large improvements in

disambiguation accuracy (Scarlini et al., 2020). As such, in contextualised models like BERT (Devlin et al., 2019) and RoBERTa (Liu et al., 2019) the ability to handle word polysemy is inherent; which has greatly contributed to the success of these models.

Polysemy in vision is a multimodal problem, as image context may present itself across various modalities. This requires that VLM understand the complex interplay between the visual and its contextual data. Interaction between modalities is particularly relevant in vision-language tasks, which often aim to generate or match textual descriptions of visual content (Baltrušaitis et al., 2017). Traditionally, works in computer vision have focused on categorisation tasks, such as object detection and classification, which were designed to be unambiguous, thereby overlooking the polysemous nature of images (Forsyth & Ponce, 2002). Central to many computer vision, and vision-language, approaches is training on human-annotated datasets like MSCOCO (Lin et al., 2015), which requires extensive manual effort. Moreover, to reduce effort and increase annotator agreement, such datasets were designed around straightforward and unambigious tasks. As a consequence, models trained on human-annotated datasets are effective for object-based descriptions, but they struggle to capture the nuance across multiple interpretations that emerge from differences in context.

Large-scale vision-language pre-training has many of the same ingredients that enabled NLP to make great improvements in dealing with word polysemy: self-supervised learning, web-scale datasets, and context (Radford et al., 2021; Kim et al., 2021; Li et al., 2021). However, a major difference between NLP and vision-language approaches is the prevalance of contrastive learning in vision-language (Radford et al., 2021; Li et al., 2021; Yu et al., 2022). The underlying assumption for contrastive vision-language learning is that the text and image express the same meaning, and can therefore be projected to the same point in latent space (Song & Soleymani, 2019). In Santurkar et al. (2022), it is shown this assumption may inhibit training when presented with captions which are not descriptive or have high variability. Santurkar et al. (2022) propose a solution that aims to reduce variability, instead we pose that this variability may simply be due to different meanings conveyed by the image. As illustrated in Figure 6, various captions may be valid for an image whilst conveying different meanings - discarding these reduces the richness from which we can learn.

## 2.2 VISION-LANGUAGE REPRESENTATION LEARNING

Prior research within vision-language representation learning can be broadly classified into two categories: earlier approaches that rely on task-specific fine-tuning of unimodal models, and more recent works that explicitly perform cross-modal training. VLM in the first category often leverage representations from pre-trained unimodal models, such as convolutional neural networks (Krizhevsky et al., 2017) trained on ImageNet (Russakovsky et al., 2015) or long shot-term memory (Hochreiter & Schmidhuber, 1997) trained on extensive text corpora (Karpathy & Fei-Fei, 2015; Agrawal et al., 2016; Anderson et al., 2018). These models tackle task-specific challenges using supervision derived from loss functions tailored to specific datasets, such as triplet loss for image-text retrieval on MSCOCO (Lin et al., 2015). While effective for these tasks, these models often struggle with generalization to different tasks (Karpathy & Fei-Fei, 2015; Agrawal et al., 2016).

More recently, the second category has gained momentum with VLM that focuses on training from large-scale datasets such as Conceptual Captions (CC3M and CC12M) (Sharma et al., 2018; Changpinyo et al., 2021), and LAION 400M and 5B (Schuhmann et al., 2021; 2022). Contrastive learning is central to this large-scale training, as it aims to optimize the similarity between matching pairs and minimize it for mismatching pairs, thereby addressing key challenges in vision-language representation learning. An observation concerning these contrastive approaches, also made by Song & Soleymani (2019), is that forcing multiple meanings to a single point can have negative influence on learning as it artificially compresses the embedding space, and reduces nuance between meanings. On datasets like MSCOCO (Lin et al., 2015) and Flickr30k (Plummer et al., 2016), this has a limited impact as their captions are highly descriptive of the image, but this does not extend to real world image-text pairs which may convey diverse meanings. As existing benchmarks are inadequate for evaluating this, it necessitates the development of datasets that address image polysemy.

A related problem to polysemy, as focused on by Song & Soleymani (2019), is that of partial matching between image and text, as in multi-view embedding (Ren et al., 2015; Li et al., 2022c) The proposed solution by Song & Soleymani (2019) focuses on learning multiple local representations and matching these to the paired text, thereby primarily addressing this partial matching problem.

Instead, we argue that polysemy may occur even when considering the same local or global views, and as such a multi-view approach does not sufficiently address this. To demonstrate the limitations of existing learning paradigms and to draw attention to the notion of image polysemy we propose a benchmark to evaluate VLM across diverse captions.

Several datasets have been proposed to test models beyond conventional tasks. For instance, the Hateful Meme challenge (Kiela et al., 2021) aims to test models on detecting hateful contents from the interaction between image and text. Similarly, Theisen et al. (2020) study memes with the aim of automatically discovering political meme genres. Whilst polysemy is found in memes, in general the text in memes is intended to be complementary to the image, thereby not fitting the frame of caption. More related to our focus, Akula et al. (2023) propose a set of vision tasks on visual metaphor understanding, which require understanding of the image and the text. Improving the ability of VLM to deal with polysemy may also aid in visual metaphor understanding, their proposed dataset has a strong focus on objects, which may obscure proper assessment of polysemy.

## 3 IMP: A BENCHMARK OF IMAGE POLYSEMY

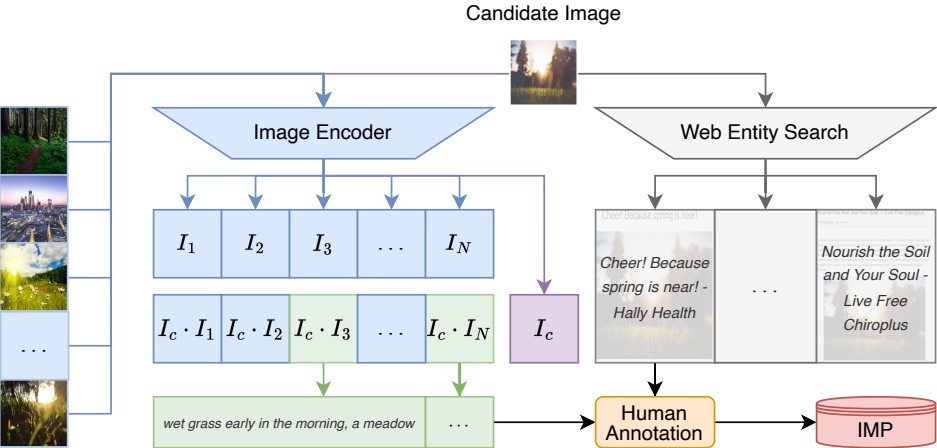

Figure 2: IMP data curation pipeline. CLIP is used as image encoder to select visually similar images from existing datasets and gather their captions; Google Vision API is used to search web-entity for each image, return the website containing the identical image and collect these website titles; Candidate captions are then cleaned, annotated, and finally five captions are selected automatically with maximal diversity for each image.

We introduce a novel benchmark for evaluating image polysemy; **IMP-5k**, that pairs images with five captions that may range from descriptive to conceptual. The pipeline for constructing the dataset is shown in Figure 2.

The images utilized for this dataset were curated from Unsplash[1], a platform renowned for its high-quality stock photography. In addition, candidate captions were gathered from two sources: existing datasets and through web curation. Captions from existing datasets were collected from visually similar images, which resulted in a set of captions (if considered relevant during annotation) that was more conceptual. CLIP-ViT-G/14 Ilharco et al. (2021) was used as the image encoder to get the embedding of the candidate image from Unsplash and images from CC3M and CC12M. We computed image-to-image cosine similarity is used to retrieve the top 3000 visually similar images for each candidate image with the similarity threshold higher than a pre-defined number (0.9 in our setting). Captions from retrieved images were used as candidate captions, and we computed text-to-text similarity as well as tokens to filter out almost identical captions. Through web curation we collected captions from websites containing identical versions of the candidate image, this allowed us to incorporate diverse real-world interpretations of the image. To ensure the quality of the gathered captions, they were subsequently checked in a cleaning and annotation process, which filtered

---

[1] https://unsplash.com/

out captions of poor quality or those which were not considered relevant to the image. From the remaining captions, we automatically selected five captions for each image by optimising for diversity across the captions. More details on the cleaning, annotation, and auto-selection can be found in Appendix A.

Table 1 shows the statistics of IMP compared to existing pre-training and fine-tuning scale datasets. IMP-5k has 25k captions selected from more than 400k valid captions with maximal diversity through clustering. The term IMP and IMP-5k will be used interchangeably in later secations. For fine-tuning, we split a set of 18k images to a 17k training set (noisyIMP-17k) with 10 captions and a 1k (noisyIMP-1k) validation set with 5 captions. Captions in noisyIMP-17k were selected based on the same rule as IMP-5k, and severe outliers (meaningless captions) were excluded by additional human check. The number of image samples in noisyIMP-17k is comparable to Flickr30k (Plummer et al., 2016), whereas the size of test split is the same as the MSCOCO test set (Lin et al., 2015). The noisyIMP-1k set is used for hyperparameter tuning and model selection. In terms of average caption length IMP is comparable to other fine-tuning datasets, while it has a higher standard deviation as IMP contains captions from existing pre-training datasets.

Table 1: Statistics of IMP compared to existing pre-training and fine-tuning datasets with official or commonly used train-test splits. MPL2D is the mean L2 distance between images and their paired captions in the embedding space, averaged over the whole dataset.

| Statistics | Pre-training | | | | Fine-tuning | | | | Benchmark |
|---|---|---|---|---|---|---|---|---|---|
| | CC3M | CC12M | SBU | RedCaps(20) | Flickr8k | Flickr30k | | MSCOCO | | IMP (ours) |
| | - | - | - | - | - | Train | Test | Train | Test | - |
| Unique images | 2.8M | 11.2M | 849k | 3.1M | 8k | 29k | 1k | 113K | **5k** | **5k** |
| Caption(s) per image | 1 | 1 | 1 | 1 | 5 | 5 | | 5 | | 5 |
| Avg caption length | $11.73 \pm 4.21$ | $19.37 \pm 15.25$ | $12.06 \pm 5.27$ | $12.14 \pm 10.63$ | $11.78 \pm 3.89$ | $11.63 \pm 3.24$ | | $10.78 \pm 3.01$ | | $11.13 \pm \mathbf{5.13}$ |
| Web curation | ✓ | ✓ | ✓ | ✓ | ✗ | ✗ | | ✗ | | ✓ |
| Human annotation | ✗ | ✗ | ✗ | ✗ | ✓ | ✓ | | ✓ | | ✓ |
| MPL2D | $1.416 \pm 0.317$ | $1.393 \pm 0.370$ | $1.263 \pm 0.026$ | $1.312 \pm 0.109$ | $1.163 \pm 0.021$ | $1.167 \pm 0.021$ | | $1.170 \pm 0.022$ | | $\mathbf{1.221 \pm 0.031}$ |

To measure the diversity among the captions, we compute the mean paired L2 distance (MPL2D) across the whole dataset. This score considers the image embedding as the cluster centroid, and denote text embeddings (from all paired captions) as points in the cluster. To reduce the bias from model's pre-training dataset, the embeddings are obtained with CLIP-ViT-B/32 (Radford et al., 2021) with four different checkpoints, and ALIGN (Jia et al., 2021). MPL2D is computed for IMP, the test set of fine-tuning datasets, and the whole pre-training datasets, for RedCaps (Desai et al., 2021) we evaluate on the year 2020 split. From obtained MPL2Ds we observe that IMP benefits from the two ways of data collection combined with manual curation, as IMP has a larger MPL2D than MSCOCO and Flickr30k which both have multiple captions for each image. Pre-training datasets can have larger MPL2D score and standard deviation, this is mainly due to noisy captions, and the fact that each image is paired with only one caption. It's worth mentioning that SBUcaptions (Ordonez et al., 2011) has lower MPL2D and much lower standard deviation than CC3M and CC12M. The main reason for this is that SBUcaptions uses Flickr queries and filtering the noisy results, thus most captions are descriptive. RedCaps (Desai et al., 2021) is a web-curated dataset where the image-text pairs are from different subreddits (tags, group spaces). RedCaps can naturally group visually unrelated images through a common semantic meaning which is the subreddits, and also allow images which share the same objects to have different captions.

For further qualitative comparison, we show four example from IMP and MSCOCO each in Figure 1, demonstrating the visual similarity between the datasets. However, when comparing the captions between these datasets we see that the captions for IMP are more diverse, incorporating highly conceptual captions such as "No care in the world" as well as descriptive captions as "Steam from a morning cup of tea or coffee". In contrast, MSCOCO features only descriptive captions, which focus strongly on the main object(s). By moving beyond pure description we enable richer exploration of captions, allowing IMP to serve as a comprehensive benchmark to evaluate the ability of VLM on image polysemy.

## 4 EXPERIMENTS

In this section, we present the results of two cross-modal experiments (zero-shot and fine-tuned) to evaluate the performance of VLM on **IMP**, and provide both quantitative and qualitative analysis. As cross-modal retrieval aims to retrieve the most relevant image given a text query, or vice versa, it allows for direct testing of the alignment between the two modalities, thus measuring the potential for contrastive learning to deal with image polysemy.

**Models.**    We categorize the VLM evaluated on IMP into three groups based on their architecture (Awais et al., 2023): dual-encoder, fusion, and other. Dual-encoder models are composed of separate encoders for image and text, the loss is computed using between the outputs of these two encoders. We evaluate the following dual-encoder VLM: CLIP (Radford et al., 2021), ALIGN (Jia et al., 2021), AltCLIP (Chen et al., 2022), ConvNeXt-CLIP (Liu et al., 2022), and ALBEF (Li et al., 2021). Fusion models have a module that combines the image and text features, in addition to the two encoders, which allows for richer pre-training tasks. We evaluate the following fusion VLM: BLIP (Li et al., 2022b), FLAVA (Singh et al., 2022), and Coca (Yu et al., 2022). Additionally, we evaluate the following other VLM: EVA-02 with encoder-decoder (Fang et al., 2023), BLIP2 (Li et al., 2023), which uses a frozen LLM, ImageBind (Girdhar et al., 2023), which uses multiple modalities of paired data. SetEmbedding (Kim et al., 2022), which uses slot attention (Locatello et al., 2020) for multi-view cross-modal retrieval.

Because of the unavailability of pre-training checkpoints and/or implementation, other state-of-the-art (SOTA) models such as Florence (Yuan et al., 2021) and FILIP (Yao et al., 2021), are not included. All checkpoints are either obtained from their official repositories (Radford et al., 2021; Ilharco et al., 2021; Wightman, 2019; Li et al., 2022a) or the HuggingFace model hub.

**Metrics.**    For evaluation, we use Recall@K (R@K) metric, which is the percentage of queries that have at least one relevant item in the top-$K$ retrieved items. We also report the RSUM following (Chen et al., 2021; Kim et al., 2022), which is the sum of Recall@K for $K = 1, 5, 10$ from both image-to-text and text-to-image retrieval tasks. Further results on Median Rank (MedR) and Mean Rank (MeanR) can be found in Appendix C.

### 4.1 ZERO-SHOT EVALUATION

We report the results of zero-shot evaluation of the VLM on IMP in Table 6. Since most of models in the table has the same or larger visual backbone than ViT-L/14, we pick CLIP with ViT-L/14 pretrained on CLIP400M (Radford et al., 2021) as the baseline for comparison. Overall, we observe incorporating additional losses (such as COCA with captioning loss) and tasks (such as FLAVA with multiple pre-training tasks) benefits the performance of the VLM, additionally model size and pretraining is a factor as well.

EVA-02-L/14 achieves the bset RSUM score and best image-to-text performance out of all models. When comparing the two retrieval tasks, we consistently see that models perform much better on image-to-text than on text-to-image, which can be explained by the VLM doing well on matching images to descriptive captions, but struggling when matching conceptual captions to images. From the highlighted results we observe that model with higher RSUM almost always have higher individual recall score, one exception is ImageBind, which has a lower image-to-text recall but higher text-to-image recall than EVA-02-L/14.

Surprisingly, BLIP2-g using the larger ViT-g/14 image encoder is outperformed by BLIP2-ViT-L with the ViT-L/14. Moreover, BLIP2-g-COCO (BLIP2 with ViT-g/14 from EVA-CLIP (Fang et al., 2023), trained on MSCOCO) performs better on image-to-text, but worse on text-to-image. These observations further highlight the importance of model size and training data. To study their effects, we compare the results of selected CLIP variants in Table 3, with more results in Appendix C.

When comparing model size, we find that VLMs with greater model size achieve better RSUM scores. This aligns with observations of CLIP on other datasets (Radford et al., 2021), demonstrating that this asymptotic nature also applies to the image polysemy setting. However, we do see observe an interaction here with the training data. For instance, in the DataComp1B (Gadre et al., 2023) setting, model performance drops significantly more when the model size decreases, indicating that

smaller models trained on DataComp1B have lower generalization ability on IMP than the same model trained on LAION2B.

Table 2: Recall@K (R@K) scores for zero-shot cross-modal retrieval on IMP. Evaluation results on both 1K test setting (average of 5-fold test dataset) and 5K test setting are presented. The best results within each recall column are highlighted with bold text. The best results within each group of models are highlighted with underline.

| | 1K Test Images | | | | | | | 5K Test Images | | | | | | |
| | Image-to-Text | | | Text-to-Image | | | RSUM | Image-to-Text | | | Text-to-Image | | | RSUM |
| Method | R@1 | R@5 | R@10 | R@1 | R@5 | R@10 | | R@1 | R@5 | R@10 | R@1 | R@5 | R@10 | |
|---|---|---|---|---|---|---|---|---|---|---|---|---|---|---|
| CLIP-L/14 | 24.4 | 51.8 | 64.5 | 15.1 | 36.9 | 48.9 | 241.6 | 11.1 | 27.3 | 37.3 | 6.0 | 17.1 | 25.1 | 124.0 |
| AltCLIP | 27.3 | 55.7 | 68.2 | 16.8 | 39.5 | 51.5 | 259.1 | 12.8 | 30.3 | 41.1 | 6.7 | 19.1 | 27.2 | 137.1 |
| ALIGN | 28.2 | 55.3 | 68.3 | 16.7 | 40.0 | 51.7 | 260.1 | 13.6 | 31.3 | 41.9 | 6.8 | 19.4 | 27.6 | 140.5 |
| CovNeXt-CLIP | 29.8 | 58.2 | 71.0 | 18.3 | 42.0 | 53.4 | 272.7 | 14.5 | 32.9 | 43.7 | 7.8 | 20.9 | 29.1 | 149.0 |
| ALBEF | 21.3 | 46.8 | 59.2 | 3.8 | 15.2 | 26.4 | 172.7 | 9.3 | 23.2 | 32.5 | 2.0 | 8.4 | 14.2 | 89.5 |
| BLIP | 23.1 | 49.5 | 62.1 | 15.4 | 36.8 | 48.4 | 235.3 | 11.2 | 25.6 | 35.1 | 6.4 | 17.4 | 24.5 | 120.3 |
| FLAVA | 26.5 | 53.8 | 66.4 | 16.7 | 38.8 | 50.6 | 252.8 | 12.3 | 28.8 | 39.2 | 6.9 | 18.5 | 26.3 | 132.0 |
| COCA | 28.5 | 57.8 | 70.6 | 16.7 | 39.9 | 51.7 | 265.2 | 13.4 | 31.7 | 43.2 | 6.7 | 19.2 | 27.3 | 141.4 |
| BLIP2-g | 23.6 | 51.6 | 65.4 | 15.2 | 38.2 | 50.7 | 244.7 | 10.1 | 26.3 | 36.7 | 5.9 | 17.3 | 25.3 | 121.5 |
| BLIP2-g-COCO | 24.4 | 51.2 | 64.2 | 14.5 | 36.6 | 48.5 | 239.5 | 11.2 | 26.9 | 37.5 | 5.6 | 16.6 | 24.2 | 122.0 |
| BLIP2-ViT-L | 28.4 | 56.3 | 69.5 | 17.2 | 40.5 | 52.8 | 264.7 | 13.0 | 31.4 | 42.0 | 6.9 | 19.4 | 27.7 | 140.5 |
| ImageBind | 29.1 | 57.46 | 70.5 | 18.8 | 42.8 | 54.4 | 273.0 | 14.0 | 32.2 | 43.0 | 7.9 | 21.2 | 29.3 | 147.7 |
| EVA-02-L/14 | 31.1 | 59.5 | 71.8 | 18.5 | 41.6 | 53.2 | 275.7 | 15.3 | 33.9 | 44.8 | 7.7 | 20.9 | 29.1 | 151.6 |

Table 3: Recall@K (R@K) scores for zero-shot cross-modal retrieval on IMP using CLIP across different ViT sizes and pre-training datasets. Evaluation results on both 1K test setting (average of 5-fold test dataset) and 5K test setting are presented.

| | | 1K Test Images | | | | | | | 5K Test Images | | | | | | |
| | | Image-to-Text | | | Text-to-Image | | | RSUM | Image-to-Text | | | Text-to-Image | | | RSUM |
| Method | Para | R@1 | R@5 | R@10 | R@1 | R@5 | R@10 | | R@1 | R@5 | R@10 | R@1 | R@5 | R@10 | |
|---|---|---|---|---|---|---|---|---|---|---|---|---|---|---|---|
| *CLIP400M*(Radford et al., 2021) | | | | | | | | | | | | | | | |
| RN50 | 102M | 25.1 | 52.6 | 65.6 | 14.9 | 36.8 | 48.6 | 243.7 | 11.0 | 27.8 | 37.1 | 6.0 | 16.9 | 24.5 | 123.3 |
| RN101 | 120M | 24.7 | 53.1 | 65.7 | 15.0 | 36.4 | 48.3 | 243.2 | 11.1 | 27.2 | 37.9 | 5.9 | 17.1 | 24.8 | 123.9 |
| B/32 | 150M | 23.5 | 51.7 | 64.5 | 15.1 | 37.2 | 49.1 | 241.2 | 10.4 | 26.7 | 36.9 | 5.9 | 17.1 | 25.0 | 122.1 |
| B/16 | 150M | 25.1 | 52.7 | 66.1 | 15.6 | 37.7 | 49.4 | 246.5 | 11.2 | 28.1 | 38.1 | 6.2 | 17.8 | 25.5 | 127.0 |
| L/14 | 428M | 24.4 | 51.8 | 64.5 | 15.1 | 36.9 | 48.9 | 241.6 | 11.1 | 27.3 | 37.3 | 6.0 | 17.1 | 25.1 | 124.0 |
| L/14-336 | 428M | 26.5 | 53.3 | 66.5 | 15.7 | 38.0 | 49.5 | 249.6 | 12.1 | 29.0 | 39.2 | 6.2 | 17.8 | 26.0 | 130.4 |
| *Laion400M* (Schuhmann et al., 2021) | | | | | | | | | | | | | | | |
| B/32 | 150M | 26.7 | 55.1 | 68.3 | 15.8 | 38.5 | 50.2 | 254.5 | 12.0 | 30.0 | 40.4 | 6.4 | 18.0 | 26.0 | 132.8 |
| B/16 | 150M | 28.0 | 56.0 | 70.1 | 16.9 | 39.7 | 51.8 | 262.6 | 13.2 | 30.5 | 41.1 | 7.2 | 18.9 | 27.3 | 138.2 |
| L/14 | 428M | 28.7 | 57.4 | 70.5 | 17.7 | 40.8 | 52.3 | 267.3 | 13.5 | 32.0 | 42.7 | 7.5 | 20.3 | 28.4 | 144.3 |
| *DataComp1B*(Gadre et al., 2023) | | | | | | | | | | | | | | | |
| B/32 | 150M | 15.4 | 37.1 | 50.3 | 9.1 | 25.1 | 35.3 | 172.4 | 5.8 | 16.9 | 24.9 | 3.3 | 10.0 | 15.7 | 76.6 |
| B/16 | 150M | 27.2 | 55.9 | 69.0 | 16.0 | 38.4 | 50.1 | 256.6 | 13.0 | 31.1 | 41.7 | 6.5 | 18.3 | 26.5 | 137.1 |
| L/14 | 428M | 29.6 | 57.9 | 71.0 | 18.5 | 41.9 | 53.3 | 272.2 | 14.2 | 32.5 | 44.1 | 7.6 | 20.8 | 29.2 | 148.4 |
| *LAION2B*(Schuhmann et al., 2022) | | | | | | | | | | | | | | | |
| B/32 | 150M | 29.0 | 59.0 | 71.5 | 17.5 | 40.7 | 52.6 | 270.2 | 14.0 | 32.1 | 43.5 | 7.0 | 20.0 | 28.2 | 144.9 |
| B/16 | 150M | 28.6 | 58.2 | 70.8 | 17.9 | 41.3 | 53.0 | 269.7 | 13.8 | 32.4 | 43.3 | 7.4 | 20.1 | 28.3 | 145.3 |
| L/14 | 428M | 29.6 | 58.0 | 70.4 | 18.4 | 42.7 | 54.3 | 273.5 | 14.2 | 32.9 | 43.9 | 7.7 | 20.9 | 29.2 | 148.7 |
| H/14 | 986M | 29.2 | 57.4 | 70.6 | 18.7 | 42.7 | 54.3 | 273.0 | 14.1 | 32.4 | 43.3 | 7.9 | 21.1 | 29.4 | 148.2 |
| g/14 | 1.36B | 30.7 | 59.4 | 72.7 | 20.0 | 44.1 | 55.8 | 282.7 | 14.8 | 33.9 | 44.7 | 8.6 | 22.3 | 30.7 | 155.0 |
| G/14 | 2.53B | 28.1 | 57.4 | 69.6 | 18.9 | 42.7 | 54.1 | 270.8 | 13.2 | 31.9 | 42.4 | 8.0 | 20.9 | 29.1 | 145.4 |

Across datasets, Laion400M (Schuhmann et al., 2021) has the same dataset size as CLIP400M (Radford et al., 2021), yet the same model trained on Laion400M performs better on IMP, this may hint at greater diversity in Laion400M. As we can observe from comparing Laion400M to Laion2B, the English language subset of LAION5B (Schuhmann et al., 2022), we obtain better performance on IMP with the same model trained on larger datasets. Moreover, due to the dataset size, Laion2B allows training of larger models, CLIP with ViT-g/14, ViT-G/14 and ViT-H/14 (Zhai et al., 2022) thus achieve higher performance than other smaller models. The performance gap between CLIP-ViT-L/14 trained on DataComp1B and Laion2B is subtle.

Despite the general behavior of performance scaling with both dataset size and model size, we find some exceptions. For instance, consider models trained on LAION2B, CLIP with ViT-L/14 performs better than models with larger ViT-H/14 and ViT-G/14 image encoders. Similarly, ViT-g/14 which is the half-precision version of ViT-G/14, has the highest performance across all models.

## 4.2 FINETUNING EVALUATION

The results of finetuning CLIP variants trained on CLIP400M are reported in Table 7, using three different methods: linear-probing, parameter-efficient fine-tuning (PEFT) (Hu et al., 2021), and full fine-tuning. We use the same hyperparameters (except for learning rate) and setup as in Radford et al. (2021) and Dong et al. (2022) for all methods. We set the base learning rate 5e-4 for linear-probing, 1e-5 for full fine-tuning, and 1e-4 for PEFT. These rates were selected using CLIP-ViT-B/32 and grid search as in Radford et al. (2021), with learning rates ranging from 1e-3 to 1e-6 and evauating on the 1k validation set.

For linear-probing, we test two commonly used strategies: finetuning the last transformer layer, which we report on in Table 7, and adding an additional linear layer ontop of the frozen CLIP model, which we report on in Appendix C. PEFT can be considered an intermediate strategy between linear-probing and fine-tuning, as it adds trainable parameters to each layer while keeping the original parameters frozen. For PEFT, we use LoRa (Hu et al., 2021) and adopt the default hyperparameters from Sourab Mangrulkar et al. (2022). Additional details can be found in Appendix A. We use triplet loss with margin 0.1 as the loss function for all methods.

Table 4: Recall@K (R@K) scores for fine-tuning cross-modal retrieval on IMP using CLIP (CLIP400M) and SetEmbedding. Evaluation results on 5K test setting are presented.

| | Linear Probing | | | | | | | Lora | | | | | | | Fully Finetune | | | | | | |
| | Image-to-Text | | | Text-to-Image | | | RSUM | Image-to-Text | | | Text-to-Image | | | RSUM | Image-to-Text | | | Text-to-Image | | | RSUM |
| Method | R@1 | R@5 | R@10 | R@1 | R@5 | R@10 | | R@1 | R@5 | R@10 | R@1 | R@5 | R@10 | | R@1 | R@5 | R@10 | R@1 | R@5 | R@10 | |
|---|---|---|---|---|---|---|---|---|---|---|---|---|---|---|---|---|---|---|---|---|---|
| RN50 | 6.2 | 19.0 | 29.8 | 4.5 | 15.3 | 24.3 | 90.1 | 9.6 | 26.9 | 39.7 | 6.3 | 20.2 | 30.9 | 133.7 | 10.6 | 28.9 | 41.6 | 6.8 | 21.4 | 32.5 | 141.8 |
| RN101 | 6.4 | 20.2 | 30.9 | 4.5 | 15.5 | 24.7 | 102.1 | 10.0 | 27.5 | 39.0 | 6.6 | 20.7 | 31.6 | 135.4 | 10.4 | 30.4 | 43.7 | 7.2 | 22.9 | 34.2 | 148.8 |
| B/32 | 10.1 | 28.7 | 42.5 | 6.8 | 22.2 | 33.8 | 144.2 | 11.8 | 30.7 | 44.2 | 7.5 | 23.6 | 35.3 | 153.1 | 12.0 | 32.9 | 45.8 | 7.9 | 24.6 | 36.6 | 159.8 |
| B/16 | 10.3 | 29.3 | 42.1 | 7.3 | 22.9 | 34.1 | 146.0 | 12.7 | 33.3 | 46.3 | 8.2 | 24.8 | 36.6 | 161.9 | 13.0 | 34.8 | 48.0 | 8.6 | 25.8 | 38.0 | 168.2 |
| L/14 | 12.3 | 31.8 | 46.1 | 8.0 | 25.1 | 37.1 | 160.5 | 14.0 | 33.8 | 46.5 | 9.0 | 26.0 | 37.8 | 167.1 | 14.7 | 36.1 | 50.3 | 9.5 | 27.6 | 39.8 | 178.1 |
| L-14-336 | 14.4 | 34.2 | 46.9 | 9.4 | 26.4 | 38.2 | 167.5 | 15.9 | 37.2 | 50.4 | 10.0 | 29.1 | 40.1 | 182.7 | 16.0 | 39.1 | 51.7 | 10.9 | 29.9 | 40.8 | 188.4 |
| SE-101 | 9.7 | 28.4 | 41.9 | 6.5 | 21.8 | 33.5 | 141.9 | - | - | - | - | - | - | - | 11.5 | 31.0 | 44.9 | 7.4 | 24.3 | 35.8 | 154.9 |
| SE/32 | 10.1 | 28.7 | 42.1 | 6.7 | 22.2 | 33.7 | 143.5 | - | - | - | - | - | - | - | 12.1 | 33.0 | 46.4 | 7.9 | 26.0 | 38.7 | 164.1 |

Overall, the performance of CLIP is improved by fine-tuning, with full fine-tuning achieving the best overall scores. Nonetheless, the performance increase of all three methods is minimal, obtaining R@K and RSUM significantly lower than zero-shot on MSCOCO. Across the finetuning settings, the results again exhibit an asytmpotic trend; increasing the model size and the number of parameters to be finetuned leads to better performance on the cross-modal retrieval task. Meanwhile, one notable exception is when performing linear probing on RN50 and RN101, where the resulting scores significantly degraded and are lower than the zero-shot scores. These observations suggest that the challenges presented by IMP go beyond being a domain shift, and demonstrate a clear limitation in existing learning paradigms.

An approach which may aid in addressing image polysemy is multi-view cross-modal retrieval as in (Song & Soleymani, 2019). To this end we implement the SOTA multi-view approach, SetEmbedding (Kim et al., 2022), with SE-101 model consists of ResNeXt-101 and BERT. Apart from the original design of SetEmbedding models, we further use CLIP-ViT-B/32 as the backbone to tests a new variant of SetEmbedding, SE/32. We choose linear-probing (finetuning the last layer of encoders along with slot attention modules), and full fine-tuning for training SetEmbedding models, as the added modules are not pre-trained. The results can be seen in lower half of Table 7.

We find that SE-101 has lower performance than CLIP-ViT-B/32 in both the linear probing and the full fine-tuning scenarios. However, SE/32 has a lower score in linear probing case but a higher score in the full fine-tuning case, resulting in text-to-image performance that is higher than CLIP-ViT-B/16. This relatively high text-to-image recall performance is different from the other VLM,

which may indicate that multi-view models are better at handling polysemy than single-view models. Nevertheless, a performance gap remains.

### 4.3 QUALITATIVE ANALYSIS

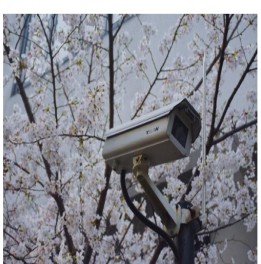 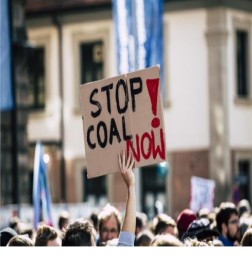 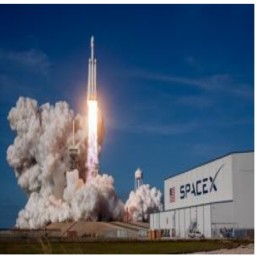 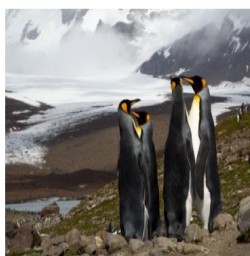

Bridge traditional cultures and modern techniques.  I guess my bill of electricity will be much higher than before.  Users of Uber may hope they can go to work in the air like this to avoid rush hour traffic  Imagine you are lost and they are trying to find out how to help you.

Figure 3: Example of hard captions in image-text matching task from IMP. "Hard" indicates the caption is still wrongly predicted as mismatching after fine-tuning.

For further analysis, we implement a simple image-text matching task where we predict whether the image and text are matched or not by passing a concatenation of the image and text embedding to a lienar layer followed by a softmax output. The pre-trained image text matching uses frozen CLIP-ViT-B/32 as feature extractor and only train the linear layer, which directly uses the cosine similarity computed by CLIP to predict. The binary cross entropy loss is added along with the triplet loss during fine-tuning, with CLIP-ViT-B/32 (unfreeze last layer) as the backbone and all other settings are the same as in the fine-tuning evaluation in Section 4.2. As we are particularly interested in determining if models can recognise whether highly conceptual captions are correctly matched to images, we focus on the false negative rate (FNR), which measures whether captions are incorrectly identified as mismatching.

The results of this analysis show that the FNR before fine-tuning on the image-text matching task is 14.6%, and after is 6.5%. As such, fine-tuning does improve the models capabilities to deal with challenging captions, but the FNR remains fairly high (as the result after fine-tuning on MSCOCO is 1.1%). We show a few examples which are still wrongly predicted after training in Figure 3, which we named them "hard captions". For instance, the second caption "I guess my bill of electricity will be much higher than before" was predicted as mismatching, while a human would likely consider this a valid caption for the image. Additional hard caption examples and analysis can be found in Appendix D.

## 5 CONCLUSION

We propose IMP, a new benchmark to challenge the capability of VLMs on image polysemy, which is the phenomenon that a single image may convey multiple different meanings. IMP consists of 23k images with diverse captions curated from the web and through manual annotation. We evaluated a wide-range of SOTA VLM models on IMP in both zero-shot and fintuning settings, and find that existing models struggle to learn from polysemous image-text pairs. Furthermore, we tested if a multi-view approach may aid in overcoming this issue, and find that it similarly struggles, but that it achieves relatively better text-to-image retrieval performance, which we regarded as crucial for understanding image polysemy. In this work we emphasised the polysemous nature of images and demonstrated how existing learning paradigms for vision-language struggle in addressing it. Our hope is that IMP can serve as a benchmark for future research on image polysemy and shape improvements in vision-language representation learning.

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

## A    IMPLEMENTATION DETAILS

All models evaluated in this work were implemented with PyTorch (Paszke et al., 2019). For fine-tuning experiments we set the max epoch to 6, batch size 128, use AdamW (Loshchilov & Hutter, 2019) optimizer with cosineannealing learning rate scheduler.

Because there is no public codebase for SetEmbedding models (Kim et al., 2022) we implement the model with (Paszke et al., 2019) v 2.0.1. We follow most setting in (Kim et al., 2022) of ResNeXt-101 + BERT, scale the learnring rate for set prediction module with 0.1. The learning rates for CNN and BERT are scaled with 0.01 and 0.1 respectively. For SE/32, the learning rates for both image encoder and text encoder are scaled with 0.5. Due to smaller dataset size comparing to MSCOCO (Lin et al., 2015), instead of training for 50 epochs, we train for 12 epochs with cosineannealing learning rate scheduler.

All zero-shot experiments are done with two NVIDIA A6000 PCIe GPUs with 48GB memory. Fine-tuning experiments are done with three NVIDIA A6000 PCIe GPUs with 48GB memory.

## B    IMP PIPELINE DETAILS

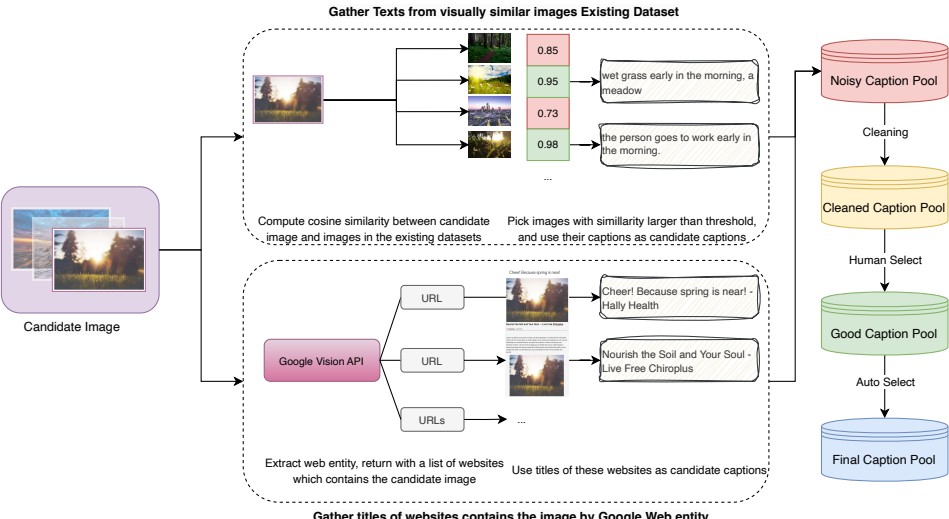

Figure 4: Detailed Pipline of the project with examples.

**Caption Gathering:**    We gather captions in two distinct ways:

1. From existing datasets: We use CLIP-ViT-G/14 (Ilharco et al., 2021) trained on LAION2B (Schuhmann et al., 2022) as image encoder. We extract image embeddings from 2M images of CC3M (Sharma et al., 2018) and a 2M subset of CC12M (Changpinyo et al., 2021), and store them for similarity search. We then compute the cosine similarity between candidate image and all stored image embeddings, and select the top 3000 images with highest similarity. We then borrow the captions from these similar images and use them as candidate captions.

2. Through web curation: Google Vision API was utilized for web-entity searches, which will return a list of websites containing identical or visually similar image, and web entities detected in the image. We only include websites with identical images, and further filter the website list by comparing between web entities detected in the image and keywords in the Unsplash dataset. This prevent visually similar images from being included in the list. We then collect the titles of these websites as candidate captions.

Captions from these two sources respectively are different in two ways:

1. The former is cleaned alt-text, while the latter is the title of the web page.

2. The former one only visually similar images can be found, while the latter one only consider identical images.

**Caption Cleaning:**  The gathered captions are often noisy, containing HTML tags, website names as suffixes or prefixes, hashtags in website titles, a high rate of word repetition, and other irrelevant information. We use packages such as ftfy to first clean the captions, and use token analysis to remove captions with high word repetition rate, which are likely to be a list of keywords curated from internet. Following Sharma et al. (2018), we also remove captions with with no noun, verb or adjective. The captions are cleaned to remove most of these noises while keeping the sentences fluent and readable. Finally, only caption which has at least 5 words are kept.

**Caption Annotation 1:**  Annotators were tasked with classifying each candidate caption as either "good" or "bad" based on detailed guidelinesto maximally capture the diversity of captions while ensuring minimal subjectivity. The quality assurance process was conducted in two stages, inter-annotator agreement of which the same set of images was annotated by multiple annotators, and review-feedback loop of which an initial set of annotations was reviewed by head annotator and provide feedback to other annotators. The inter-annotator agreement stage was used to ensure that the annotators were consistent in their judgments. The review-feedback loop was used to ensure that the annotators were following the guidelines and to provide feedback on their annotations.

**Caption Annotation 2:**  One annotator used a score-based measuring standard, which consisted of three metric, each ranging from 1-5:

1. Descriptive Score: Measures the quality of a descriptive caption. Higher scores are given to more accurate and detailed captions, while lower scores are assigned to inaccurate or non-informative captions.

2. Conceptual Score: Assesses the quality of a conceptual caption. Captions that capture more themes receive higher scores, while overly generic captions receive lower scores.

3. Additional Score: Evaluates the additional qualities of a caption, such as emotional impact, uniqueness, and informativeness. The richer the caption in these aspects, the higher the score.

In the score-based standard, a caption is considered good if it has either a high Descriptive Score or a high Conceptual Score. This standard enables more fine-grained evaluation of the captions. The results from annotation 2 also joins the feedback-loop of annotation 1, and the final decision is made by the head annotator. We acknowledge that the process of classifying captions can be inherently subjective. However, the use of detailed guidelines, multiple annotators, and a rigorous quality assurance process aimed to mitigate this subjectivity to the extent possible.

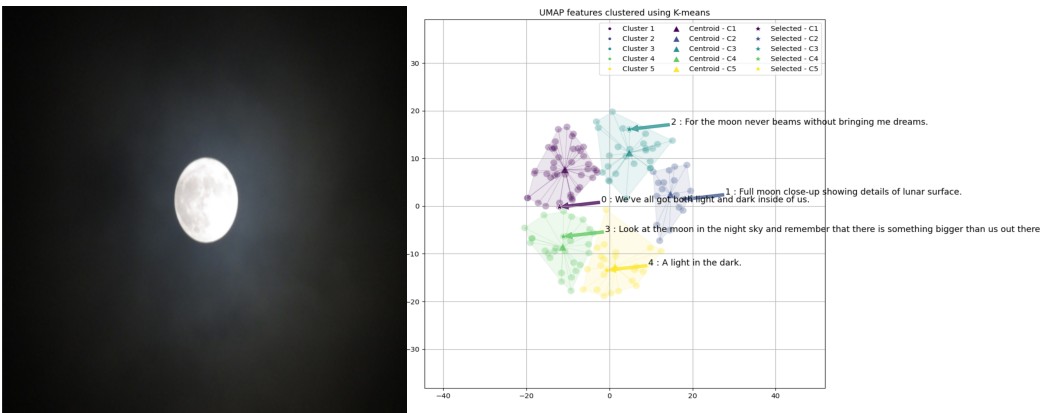

Figure 5: An example of using Sentence-BERT, UMAP, and K-means to select diverse captions.

**Diverse Caption Selection:** After the manual selection process, we were left with a pool of good captions for each image. Any of captions in the pool of an image can be included in the final dataset. To ensure diversity, we employed Sentence-BERT (Reimers & Gurevych, 2019; Thakur et al., 2021) to extract textual features from the captions, as it is adept at capturing semantic similarities between sentences. We opted not to use CLIP for this task because it is more focused on aligning image and text features, rather than capturing the nuanced semantic relationships between different text snippets. We then project the textual features into 2D space using UMAP (McInnes et al., 2020), and use K-means clustering to group the captions into five clusters. One caption was randomly selected from each cluster to create a diverse set of five captions for each image.

**Quality Control:** Image-caption pair in IMP-5k has all passed human-evaluation and agreed by annotators. We have verified examples for the diverse caption selection process for IMP-5k, annotators agreed that captions are And for noisyIMP-17k and 1k, we exclude extreme outliers in the sentenceBERT embedding space, and checked through examples selected by clustering.

## C  ADDITIONAL RESULTS

Table 5: Recall@K (R@K) scores for zero-shot cross-modal retrieval on IMP (upper). Evaluation results on both 1K test setting (average of 5-fold test dataset) and 5K test setting are presented. Linear probing with adding one additional layer on top of CLIP for cross-modal retrieval on IMP) lower. Evaluation results on 5k test setting are reported

| | 1K Test Images | | | | | | | 5K Test Images | | | | | | |
| | Image-to-Text | | | Text-to-Image | | | | Image-to-Text | | | Text-to-Image | | | |
| Method | R@1 | R@5 | R@10 | R@1 | R@5 | R@10 | RSUM | R@1 | R@5 | R@10 | R@1 | R@5 | R@10 | RSUM |
|---|---|---|---|---|---|---|---|---|---|---|---|---|---|---|
| RN50-YFCC15M | 17.7 | 41.3 | 53.1 | 10.6 | 28.5 | 38.9 | 190.2 | 7.2 | 19.5 | 28.1 | 3.9 | 12.2 | 18.3 | 89.2 |
| RN50-cc12m | 19.8 | 44.6 | 57.2 | 12.3 | 31.5 | 42.7 | 208.0 | 8.1 | 20.8 | 29.9 | 4.6 | 13.8 | 20.2 | 97.4 |
| RN50-QuickGELU-openai | 25.1 | 52.6 | 65.6 | 14.9 | 36.8 | 48.6 | 243.7 | 11.0 | 27.8 | 37.1 | 6.0 | 16.9 | 24.5 | 123.3 |
| RN50-QuickGELU-YFCC15M | 16.8 | 39.7 | 52.5 | 10.8 | 29.0 | 39.4 | 188.1 | 6.6 | 18.3 | 26.7 | 4.0 | 12.2 | 18.5 | 86.3 |
| RN50-QuickGELU-cc12m | 19.4 | 45.3 | 57.8 | 12.5 | 31.7 | 43.2 | 209.8 | 7.9 | 21.0 | 30.4 | 4.6 | 14.0 | 20.4 | 98.3 |
| RN50x4-openai | 26.0 | 53.5 | 67.0 | 15.3 | 36.7 | 48.9 | 247.4 | 11.3 | 28.9 | 39.0 | 6.3 | 17.5 | 25.2 | 128.2 |
| RN50x16-openai | 25.2 | 54.3 | 66.8 | 15.4 | 37.6 | 49.4 | 248.7 | 11.3 | 28.2 | 39.2 | 6.2 | 17.5 | 25.4 | 127.8 |
| RN50x64-openai | 25.1 | 52.9 | 66.4 | 15.6 | 37.3 | 48.9 | 246.2 | 11.3 | 28.0 | 38.0 | 6.2 | 17.4 | 25.4 | 126.3 |
| RN101-YFCC15M | 18.5 | 42.9 | 55.7 | 11.1 | 29.2 | 40.1 | 197.4 | 7.5 | 20.2 | 29.2 | 4.1 | 12.6 | 18.8 | 92.4 |
| RN101-QuickGELU-openai | 24.7 | 53.1 | 65.7 | 15.0 | 36.4 | 48.3 | 243.2 | 11.1 | 27.2 | 37.9 | 5.9 | 17.1 | 24.8 | 123.9 |
| RN101-QuickGELU-YFCC15M | 18.0 | 41.4 | 54.1 | 11.1 | 29.5 | 39.9 | 194.0 | 6.9 | 19.7 | 28.2 | 4.2 | 12.7 | 18.9 | 90.5 |
| roberta-ViT-B-32-LAION2B | 29.6 | 57.8 | 70.6 | 17.4 | 40.2 | 52.0 | 267.6 | 14.0 | 32.6 | 43.0 | 7.1 | 19.5 | 28.0 | 144.0 |
| ViT-B-16-plus-240-LAION400M | 28.8 | 57.0 | 70.7 | 17.0 | 40.4 | 52.0 | 265.9 | 13.7 | 32.2 | 42.1 | 7.1 | 19.4 | 27.8 | 142.2 |
| ViT-B-32-commonpool-m-clip | 15.8 | 38.4 | 50.6 | 8.8 | 25.3 | 36.1 | 175.0 | 5.7 | 17.2 | 25.0 | 3.0 | 9.9 | 15.5 | 76.2 |
| ViT-B-32-CommonPool-m-LAION | 14.5 | 36.5 | 49.7 | 8.5 | 24.7 | 35.0 | 168.8 | 5.0 | 15.6 | 22.8 | 2.9 | 9.5 | 15.0 | 70.7 |
| ViT-B-32-CommonPool-m-image | 16.7 | 39.6 | 53.0 | 9.4 | 27.0 | 37.8 | 183.6 | 6.2 | 17.7 | 25.7 | 3.2 | 10.6 | 16.7 | 80.0 |
| ViT-B-32-CommonPool-m-text | 16.2 | 38.0 | 51.2 | 8.9 | 25.3 | 35.9 | 175.4 | 5.6 | 17.2 | 25.3 | 3.2 | 10.0 | 15.8 | 77.1 |
| ViT-B-32-CommonPool-m-basic | 14.8 | 36.4 | 49.8 | 8.6 | 24.8 | 35.5 | 169.7 | 5.3 | 16.2 | 23.6 | 2.7 | 9.5 | 15.0 | 72.3 |
| ViT-B-32-CommonPool-m | 12.0 | 30.9 | 42.9 | 6.9 | 21.4 | 31.3 | 145.4 | 3.9 | 12.7 | 19.1 | 2.2 | 7.8 | 12.5 | 58.2 |
| ViT-B-16-CommonPool-l-clip | 26.0 | 54.9 | 67.7 | 14.7 | 36.7 | 48.5 | 248.4 | 11.6 | 28.8 | 39.7 | 5.7 | 17.0 | 24.8 | 127.7 |
| ViT-B-16-CommonPool-l-LAION | 27.8 | 56.0 | 69.1 | 15.9 | 39.3 | 50.9 | 259.0 | 12.3 | 30.3 | 41.3 | 6.3 | 18.3 | 26.5 | 135.0 |
| ViT-B-16-CommonPool-l-image | 26.1 | 54.1 | 67.5 | 15.6 | 38.0 | 49.7 | 251.1 | 11.8 | 29.3 | 39.4 | 6.1 | 17.5 | 25.7 | 129.7 |
| ViT-B-16-CommonPool-l-text | 26.0 | 52.9 | 66.7 | 14.5 | 36.2 | 48.2 | 244.5 | 11.5 | 28.3 | 39.0 | 5.7 | 16.8 | 24.5 | 125.7 |
| ViT-B-16-CommonPool-l-basic | 25.2 | 52.9 | 65.8 | 14.9 | 36.4 | 48.0 | 243.0 | 11.0 | 28.3 | 38.4 | 5.7 | 16.7 | 24.2 | 124.3 |
| ViT-B-16-CommonPool-l | 23.3 | 50.0 | 63.7 | 13.4 | 34.7 | 46.7 | 231.8 | 10.1 | 25.8 | 35.6 | 5.1 | 15.3 | 22.8 | 114.7 |
| ViT-L-14-CommonPool-xl-clip | 29.7 | 58.9 | 70.6 | 18.0 | 41.4 | 53.5 | 272.0 | 14.5 | 32.4 | 43.5 | 7.5 | 20.5 | 28.8 | 147.1 |
| ViT-L-14-CommonPool-xl-LAION | 30.0 | 58.3 | 70.7 | 18.6 | 42.7 | 54.4 | 274.7 | 14.4 | 33.1 | 43.2 | 7.9 | 21.1 | 29.2 | 149.0 |
| ViT-L-14-CommonPool-xl | 27.8 | 55.9 | 69.5 | 16.4 | 38.9 | 50.2 | 258.7 | 13.7 | 31.5 | 41.4 | 6.7 | 18.5 | 26.5 | 138.4 |
| ConvNeXt-base-LAION400M | 27.2 | 55.9 | 68.9 | 16.6 | 39.0 | 50.8 | 258.5 | 12.6 | 30.0 | 40.8 | 6.7 | 18.8 | 26.9 | 135.9 |
| ConvNeXt-base-w-LAION2B | 29.5 | 59.0 | 71.8 | 17.9 | 41.3 | 53.0 | 272.5 | 14.2 | 32.4 | 43.8 | 7.4 | 20.5 | 28.7 | 147.1 |
| ConvNeXt-base-w-LAION | 29.4 | 58.8 | 71.2 | 18.4 | 41.8 | 53.0 | 272.8 | 14.1 | 32.7 | 43.8 | 7.7 | 20.7 | 29.0 | 148.0 |
| ConvNeXt-base-w-320-LAION | 28.8 | 57.1 | 69.8 | 18.2 | 41.3 | 52.9 | 268.1 | 13.5 | 31.8 | 42.5 | 7.4 | 20.4 | 28.4 | 144.0 |
| ConvNeXt-large-d-320-LAION2B | 30.3 | 59.1 | 71.3 | 18.6 | 42.2 | 53.8 | 275.4 | 14.7 | 33.5 | 44.8 | 7.8 | 21.0 | 29.4 | 151.1 |
| ConvNeXt-xxlarge-LAION2B | 28.8 | 58.3 | 70.7 | 18.5 | 41.5 | 53.4 | 271.2 | 14.4 | 32.3 | 43.4 | 7.8 | 20.8 | 28.9 | 147.5 |
| EVA01-g-14-LAION400M | 29.9 | 58.6 | 70.7 | 18.5 | 41.9 | 53.3 | 272.8 | 14.7 | 33.0 | 43.8 | 7.7 | 20.8 | 29.4 | 149.3 |
| EVA01-g-14-plus-Merged2B | 32.5 | 60.6 | 72.3 | 18.5 | 42.2 | 54.0 | 280.1 | 15.8 | 35.4 | 46.1 | 8.1 | 21.0 | 29.5 | 155.8 |
| EVA02-B-16-Merged2B | 30.4 | 58.8 | 71.7 | 18.2 | 41.0 | 53.1 | 273.2 | 14.5 | 33.3 | 44.5 | 7.5 | 20.5 | 28.6 | 148.9 |
| EVA02-L-14-Merged2B | 30.9 | 59.4 | 72.3 | 18.5 | 41.7 | 53.3 | 276.1 | 14.9 | 34.5 | 44.9 | 7.7 | 20.9 | 29.0 | 151.9 |
| EVA02-L-14-336-Merged2B | 30.1 | 58.8 | 71.3 | 18.0 | 41.0 | 52.7 | 271.9 | 15.0 | 33.4 | 44.5 | 7.5 | 20.3 | 28.5 | 149.1 |
| RN50 | - | - | - | - | - | - | - | 3.4 | 13.5 | 21.2 | 3.0 | 10.9 | 17.5 | 69.5 |
| RN101 | - | - | - | - | - | - | - | 3.3 | 13.7 | 21.6 | 3.3 | 11.3 | 18.0 | 71.2 |
| B/32 | - | - | - | - | - | - | - | 9.4 | 27.0 | 38.4 | 6.1 | 19.6 | 29.6 | 130.1 |
| B/16 | - | - | - | - | - | - | - | 11.3 | 29.4 | 40.4 | 6.6 | 19.5 | 28.8 | 136.0 |
| L/14 | - | - | - | - | - | - | - | 12.0 | 29.4 | 42.1 | 6.5 | 19.7 | 28.9 | 138.6 |
| L/14-336 | - | - | - | - | - | - | - | 13.0 | 31.6 | 44.1 | 7.1 | 20.7 | 30.1 | 146.7 |

yfcc15m is a subset of the Yahoo Flickr Creative Commons 100M dataset (Thomee et al., 2016) which contains 15M images. QuickGELU activation function is used in the original CLIP paper

(Radford et al., 2021). Commonpool is a dataset of 12.8B image-text pairs collected from Common Craw introduced in (Gadre et al., 2023). Merged-2B is a dataset used in EVA-02 (Fang et al., 2023).

Table 6: MedR and MeanR scores for zero-shot cross-modal retrieval on IMP. Evaluation results 5K test setting are presented. The best results within each recall column are highlighted with bold text. The best results within each group of models are highlighted with underline.

| | 5K Test Images | | | |
|---|---|---|---|---|
| | Image-to-Text | | Text-to-Image | |
| Method | MedR | MeanR | MedR | MeanR |
| CLIP-L/14 | 22 | 1200 | 169 | 1301 |
| AltCLIP | 17 | 1003 | 74 | 1219 |
| ALIGN | 17 | 984 | 73 | 1224 |
| CovNeXt-CLIP | 14 | 934 | 59 | 1189 |
| ALBEF | 29 | 1773 | 554 | 1430 |
| BLIP | 25 | 1483 | 429 | 1381 |
| FLAVA | 19 | 1148 | 161 | 1298 |
| COCA | 16 | 918 | 64 | 1202 |
| BLIP2-g | 22 | 1124 | 88 | 1248 |
| BLIP2-g-COCO | 21 | 1190 | 130 | 1288 |
| BLIP2-ViT-L | 16 | 874 | 58 | 1189 |
| ImageBind | 15 | 969 | 75 | 1216 |
| EVA-02-L/14 | 14 | 734 | 51 | 1150 |

Table 7: MedR and MeanR scores for fine-tuning cross-modal retrieval on IMP using CLIP (CLIP400M) and SetEmbedding. Evaluation results on 5K test setting are presented.

| | Linear Probing | | | | Lora | | | | Fully Finetune | | | |
|---|---|---|---|---|---|---|---|---|---|---|---|---|
| | Image-to-Text | | Text-to-Image | | Image-to-Text | | Text-to-Image | | Image-to-Text | | Text-to-Image | |
| Method | MedR | MeanR | MedR | MeanR | MedR | MeanR | MedR | MeanR | MedR | MeanR | MedR | MeanR |
| RN50 | 19 | 601 | 37 | 971 | 25 | 541 | 33 | 863 | 15 | 404 | 23 | 805 |
| RN101 | 19 | 579 | 31 | 863 | 16 | 522 | 29 | 855 | 16 | 421 | 24 | 832 |
| B/32 | 18 | 586 | 35 | 861 | 13 | 277 | 21 | 756 | 13 | 296 | 20 | 756 |
| B/16 | 18 | 530 | 35 | 815 | 12 | 186 | 20 | 725 | 12 | 206 | 19 | 702 |
| L/14 | 15 | 311 | 27 | 741 | 13 | 184 | 18 | 655 | 11 | 175 | 18 | 669 |
| L-14-336 | 14 | 351 | 25 | 710 | 11 | 179 | 17 | 634 | 11 | 135 | 17 | 651 |
| SE-101 | 15 | 616 | 29 | 775 | - | - | - | - | 12 | 107 | 15 | 531 |
| SE/32 | 13 | 613 | 28 | 757 | - | - | - | - | 11 | 112 | 14 | 527 |

We reported that the MedR and MeanR are high for zero-shot results and after fine-tuning, both Medr and Meanr dropped. One thing to notice is that for Text-to-Image retrieval, MedR varies a lot with different models. Another point is that for fully-finetuned SE-101 and SE/32, even though the results on RSUM is lower than CLIP-L/14-336, they achieved lower MedR and MeanR in both Image-to-Text and Text-to-Image retrieval task. This indicates that SE model tries to learn from different caption embeddings and link them with the same image embedding, demonstrating the benefit of multiple embeddings to solve the polysemy challenge.

# D  ADDITIONAL QUALITATIVE RESULTS

We further show more examples from IMP.

In Figure 7 we use Uniform Manifold Approximation Projection (UMAP) (McInnes et al., 2020) to project high dimentional embedding into 2D space but keep the relative distance and structure. Image and text embeddings are extracted by CLIP-ViT-B/32 before and after image-text matching training, and the UMAP projection is computed on the test set. The third example from Figure 3 is stated in the plot, where the hard text means the caption which has false negative prediction. The embedding of hard text swaped to the other side of the image, which is due to image-text matching and triplet loss fine-tuning. The model tries to pull the hard text "Users of Uber may hope they can go to work in the air like this to avoid rush hour traffic" closer to the image, but the concept in the

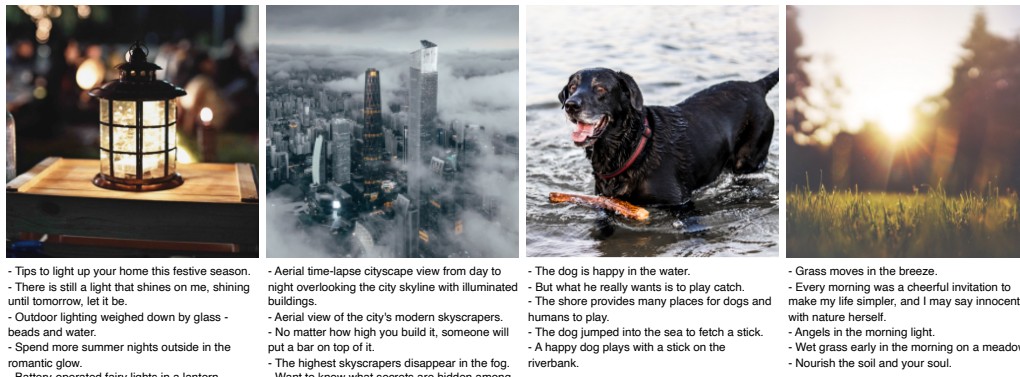

- Tips to light up your home this festive season.
- There is still a light that shines on me, shining until tomorrow, let it be.
- Outdoor lighting weighed down by glass - beads and water.
- Spend more summer nights outside in the romantic glow.
- Battery-operated fairy lights in a lantern.

- Aerial time-lapse cityscape view from day to night overlooking the city skyline with illuminated buildings.
- Aerial view of the city's modern skyscrapers.
- No matter how high you build it, someone will put a bar on top of it.
- The highest skyscrapers disappear in the fog.
- Want to know what secrets are hidden among the steel, glass, and cloud-soaring depths?

- The dog is happy in the water.
- But what he really wants is to play catch.
- The shore provides many places for dogs and humans to play.
- The dog jumped into the sea to fetch a stick.
- A happy dog plays with a stick on the riverbank.

- Grass moves in the breeze.
- Every morning was a cheerful invitation to make my life simpler, and I may say innocent, with nature herself.
- Angels in the morning light.
- Wet grass early in the morning on a meadow.
- Nourish the soil and your soul.

Figure 6: Our benchmark IMP contains images paired with five captions that range from descriptive to conceptual. This diversity among captions allows for multiple image meanings, thereby highlighting the polysemic nature of images and presenting a new challenge for vision-language models.

text is further pushed away in another direction by other training samples, thus it is not able to match with the image.

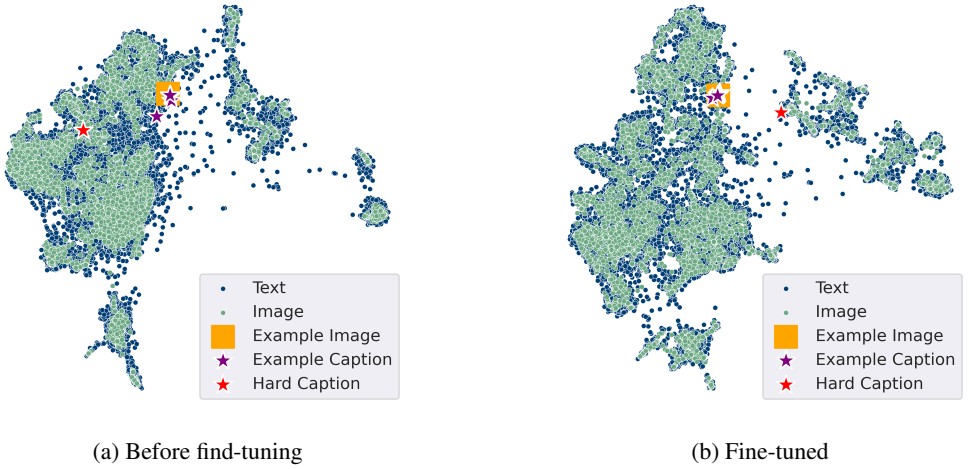

(a) Before find-tuning  (b) Fine-tuned

Figure 7: UMAP projection of the test set before and after fine-tuning, hard text is the caption which failed to match with the image.

We also show more examples of hard caption in Figure 8.

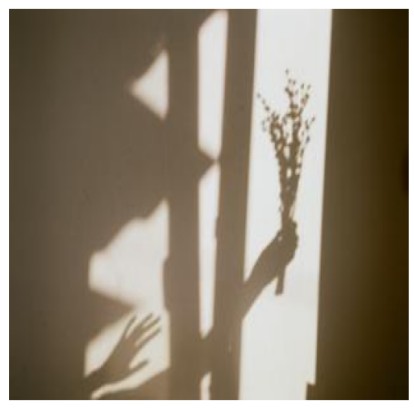

Most people think happiness is about gaining something, but it's not.

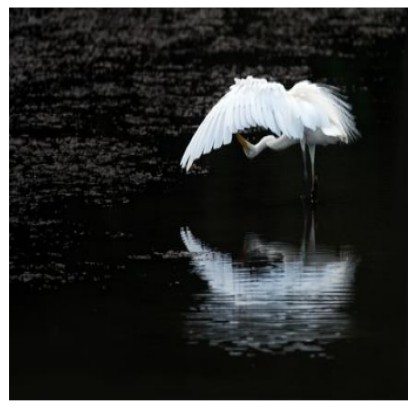

Ballet at the emperor's palace of its own, seen from below.

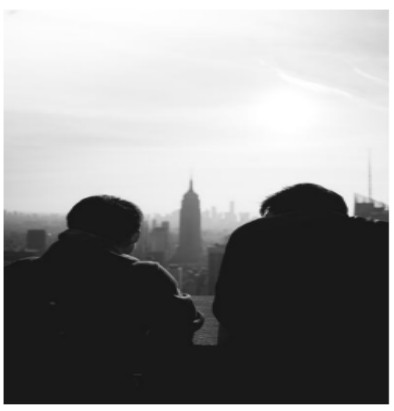

Young tourist takes a photo on his tablet PC from strengers back and watching the horizon from far

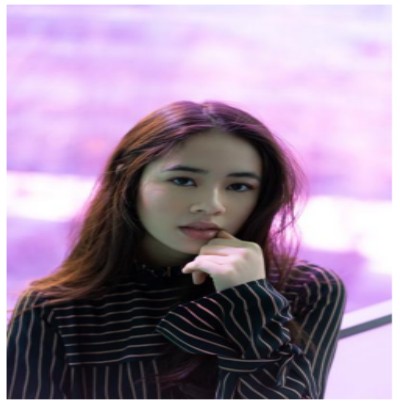

Score faces according to professional patterns as part of a plastic surgeons contest.

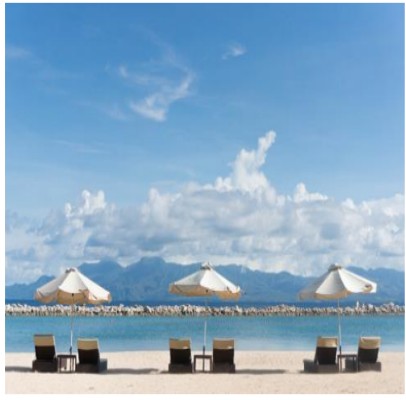

A pair of lovers will waste one seat, but what if they have a baby

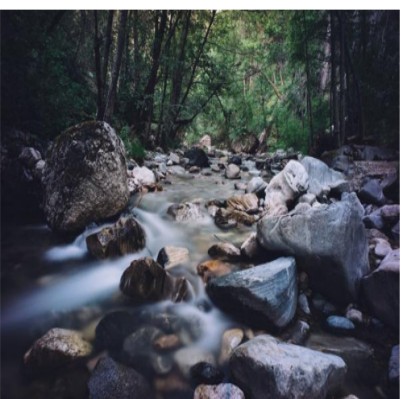

Life and death are just another false dichotomy, things floated away in front of you will never be the same.

Figure 8: More exampe of hard captions.

