# OpenReview forum: "IMP: Benchmarking Image Polysemy in Vision-Language Models"
_ICLR.cc/2024/Conference — Submitted to ICLR 2024_

### Official Review · Reviewer_VqT9 · 2023-10-31

**Soundness:** 3 good
**Presentation:** 3 good
**Contribution:** 3 good
**Rating:** 3
**Confidence:** 4

**Summary:**

This work proposes a benchmark, IMP, to evaluate understanding of image polysemy in vision-language models. The IMP benchmark includes diverse captions that range from descriptive to conceptual. A semi-automatic pipeline was constructed to select the captions, with use of human annotators. A variety of existing vision-language models are benchmarked on the IMP dataset.

**Strengths:**

The dataset is novel, and is the first to study image polysemy in the context of image-text matching / retrieval. The experiment evaluation covers a wide range of models and is sufficient.

**Weaknesses:**

I am concerned about the difficulty of the task. "Vanilla" image-text retrieval itself is not straightforward to evaluate, given the high rate of false negative caused by plausible but unrecorded matches [1,2]. The subjectivity of image polysemy amplifies all these difficulties. I wonder to what degree this task is even _possible_ to evaluate objectively. For example, it was not obvious to me that the caption and images in Fig 4. Col1 & Col 4 go together. Similarly for many of pairs in Fig 3.

The subjectiveness of this benchmark calls for a human accuracy check, given that this benchmark is so heavily based on human judgements of image meaning. The human judgements of similarity can then be used to create similarity judgements similar to [1,2] or correct false negatives. Given the dataset construction technique, I'm skeptical whether the evaluations are meaningful given that the captions are abstract enough that many of the might match plausibly match to other images. This could be corrected by verifying human accuracy on the dataset and modifying the benchmark so that correlation with human accuracy is being assessed instead.

Note that the annotation process _does not_ guarantee this, since the annotation procedure does not exclude the possibility that there are 10-15 other captions somewhere in the dataset that describe the image equally well, and vice versa for any image. You could do this by using CLIP to rank _all_ the captions in the dataset w.r.t to a particular image (and vice versa), and having humans rerank the top 50 from each cluster. This could be used to measure both human accuracy, and capture more plausible measurements of human similarity.


[1] ECCV Caption: Correcting False Negatives by Collecting Machine-and-Human-verified Image-Caption Associations for MS-COCO
[2]  Crisscrossed captions: Extended intramodal and intermodal semantic similarity judgments for ms-coco

**Questions:**

Please see the weaknesses section.

---

> ### Author Response · Authors · 2023-11-21
> **Response to Reviewer VqT9 (R5)**
>
> We thank the reviewer $\textcolor{purple}{R5}$ for insightful comments and valuable suggestions on the subjectiveness of IMP.
>
> ## Task difficulty and false negatives
>
> We thank the reviewer $\textcolor{purple}{R5}$ for starting the discussion on the difficulty of the task and false negatives as this goes directly to the core of the issue we are studying in this paper. Indeed, considering image polysemy amplifies the difficulties concerning evaluation, yet we argue that when dealing with real-world examples these difficulties are unavoidable.
>
> Increasingly datasets are collected by webscraping image-text pairs, and inherently these will contain pairs with varying levels of 'subjectivity' - some image-text pairs may exist solely because a single person decided to pair them. As such, even what is considered a 'true positive' is in the eye of the beholder (as exemplified by the reviewer's comments on the pairings in Fig 3 & 4). However, as humans we can deal with such ambiguity - or perhaps better, with such polysemy - yet, vision-language models trained with a contrastive loss presumably cannot.
>
> In our paper we test to what extent VLM trained with contrastive loss cannot deal with the challenge of polysemy, by proposing a dataset that makes explicit the presence of polysemy. Since IMP-5k is human annotated we have reliable ground truth for the true positives, making it possible to get better insight into the polysemy challenge. However, as pointed out by the reviewer we do not have annotations for true negatives - and whilst the procedure specified by the reviewer is viable we must also recognise that this would be a very costly exercise.
>
> We strongly believe that even with the existing setup, IMP dataset provides a useful testbed for getting a better understanding of image polysemy in VLM. While this understanding may at this early stage of research into image polysemy in VLM be somewhat coarse, we do believe that any model that explicitly handles image polysemy would perform significantly better on IMP.
>
> >   the captions are abstract enough that many of the might match plausibly match to other images
>
> During annotation we aimed to excluded captions which are very non-specific.
>
> For example:
> “I took this photo with my tablet.” -- Adds no contextual value to the image and can basically used as caption for any image in this dataset.
>
> The annotators are also instructed to avoid “over-thinking”. Annotators will first see the caption and then the image. The judgement is based on their first and second impression (directly finding the link or after a few seconds of thinking) of whether the caption and image can be a good pair. We find that this instruction helps reduce far-fetched pairings, but leaves the possibility for creativity and highly polsemic pairings.

---

> > ### Comment · Reviewer_VqT9 · 2023-11-23
> >
> > Thank you for your response. It does not address my main question.
> >
> > The annotation and evaluation procedures are asymmetric. The annotator sees a small set of $n_1$ viable captions only. The model being evaluated sees $n_2>> n_1$ captions, many of which could also match the image.
> >
> > The evaluations in the paper would make sense if the evaluation was done one-by-one (use a model with an image-text matching head like BLIP-1/OFA/BLIP-2 or CLIP with a threshold for match / no match) for each image and human-annotated true positive caption.
> >
> > The evaluations as they are currently done do not exclude the possibility that the model understands polysemy perfectly well and it is simply digging up other captions in the dataset that could plausibly match the image, especially since many of the captions are so abstract. It's possible that humans would even agree with the other captions the model selected.
> >
> > It's difficult to draw any claims about the ability of a model to understand polysemy, given the problem above.
> >
> > I will keep my rating, but I emphasize that I think the dataset is interesting. But you need a better evaluation protocol, or a comparison to humans under the current protocol.

---

> > > ### Author Response · Authors · 2023-11-23
> > > **Response to Reviewer VqT9 (R5)**
> > >
> > > > Thank you for your response. It does not address my main question.
> > >
> > > We thank reviewer $\textcolor{purple}{R5}$ for the further clarification of their concerns and question and we will address this point more specifically in the following.
> > >
> > > > The annotation and evaluation procedures are asymmetric.
> > >
> > > Indeed, the retrieval experiments as evaluated with R@1-10 leave open the possibility that there are unrecorded matches which are as valid as the annotated matches. However, we believe that the additional results (in the shared response and Table 7 in the updated PDF) using median and mean rank do shed some light on this. As compared to MSCOCO, the median rank is notably higher - which can still be due to the same issue, but even after fine-tuning the mean rank on both Image-to-Text and Text-to-Image is very high, which would at least imply that the models are struggling to rank correct matches towards the top.
> > >
> > > Moreover, while from an image polysemy perspective all captions are equally valid, in contrastive learning it would be much easier to closely align images and captions that express many of the same concepts. Therefore, an easy shortcut for the model would be to focus on descriptive captions and match these closely to the image - yet the higher medianR implies that even this is not happening without difficulty, and that the presence of abstract or conceptual captions is making the alignment more difficult across the board.
> > >
> > > > The evaluations in the paper would make sense if the evaluation was done one-by-one
> > >
> > > We thank the reviewer for this suggestion, we believe that in Section 4.3 we report results of exactly such an image-text matching task. We train a linear layer with binary cross entropy ontop of image-text pairs to predict whether they are matches or not. In the paper we focus on the false negative rate as we believe this most clearly demonstrates the models' ability to match highly conceptual/abstract captions. As any false negative indicates a pairing which a human annotator considered valid, but the model does not.
> > >
> > > From these results we see that the False Negative Rate (FNR) is fairly high, 14.6% before fine-tuning and still 6.5% after. As compared to 1.1% FNR on MSCOCO. This shows that there are a considerable number of captions which the models do not 'understand', thereby highlighting the challenge of polysemy.
> > >
> > > Figure 3 shows examples of four such pairings that the model was unable to correctly match (i.e., false negatives), even after fine-tuning. While we recognise that the reviewer has pointed out that these are not obvious matchings (as indeed the models struggle with them too), we do believe that it is feasible to consider that another human might match these images and captions. Moreover, for the images and captions in Figure 3 we believe that any other permutation of these images and captions would be much less obvious to a human annotator (as they are to us).
> > >
> > > For completeness we report the full results of the one-by-one image-text matching task below, and we will include these in the appendix of the paper:
> > >
> > > | Last-Layer |                         |   Predicted   |   Condition   |
> > > | ---------- |:----------------------- |:-------------:|:-------------:|
> > > |            | Total = P + N = 50,000  | Positive (PP) | Negative (PN) |
> > > | Actual     | Positive ( P ) = 25,000 |    21,357     |     3,643     |
> > > | Condition  | Negative (N)  = 25,000  |     6,624     |    18,376     |
> > >
> > > Here each image has 5 positive, which are its paired captions based on human annotation, and 5 negatives which are captions from other images (sampled captions can be from the same image without repetition).
> > >
> > > | Fine-Tuning |                         |   Predicted   |   Condition   |
> > > | ----------- |:----------------------- |:-------------:|:-------------:|
> > > |             | Total = P + N = 50,000  | Positive (PP) | Negative (PN) |
> > > | Actual      | Positive ( P ) = 25,000 |    23,389     |     1,611     |
> > > | Condition   | Negative (N)  = 25,000  |     4,777     |    20,223     |
> > >
> > > We report the below table for comparison:
> > >
> > > |       Metrics        | Last-Layer (%) | Fine-Tuning (%) |
> > > |:--------------------:|:--------------:|:---------------:|
> > > |      Precision       |     76.33      |      83.04      |
> > > |        Recall        |     85.43      |      93.56      |
> > > |       F1 Score       |     80.62      |      87.98      |
> > > | False Positive Rate  |     26.50      |      19.11      |
> > > | False Negative Rate  |     14.57      |      6.44       |

---

### Official Review · Reviewer_KVAJ · 2023-10-31

**Soundness:** 2 fair
**Presentation:** 3 good
**Contribution:** 2 fair
**Rating:** 5
**Confidence:** 2

**Summary:**

The following work presents a dataset for benchmarking polysemy in text-image pairings. Examples of difficult pairings include meme-like pairings that require drawing higher level and more abstract connections between the text and image content. Dataset curation uses a combination of google vision API and CLIP image embeddings to collect website titles as potential captions, combined with candidate captions from existing datasets. Annotator's select captions either based on descriptiveness or level of conceptual match, the latter being a more abstract linking. Diversity is encouraged by clustering captions candidates by their sentenceBERT embeddings, then selecting 1 caption candidate per cluster. Existing pretrained models are then benchmarked on this dataset through both zero-shot testing and fine-tuning.

**Strengths:**

- Tackles an important barrier in higher level image-text comprehension
- Contributes another fairly large scale dataset that can be of use to community

**Weaknesses:**

- The claim for having higher diversity in captions in this work is slightly problematic because diversity is never clearly defined. Based on their method, it seems to refer specifically to whatever euclidean distance can be associated with in sentenceBERT's embedding representation, which I am not entirely sure how to interpret.
- Some concerns regarding methodology which I elaborate on in the next section.
- While this work certainly provides a useful benchmark for polysemy, I don't think the contributions of this work sufficiently shed light on any new aspects of the problem that the community was already aware of.

**Questions:**

- The MPL2D metric measures the mean euclidean distance between image and caption embeddings, meant to capture the semantic diversity to justify the aforementioned diversity claims. There may be some concerns regarding this formulation which I would like to give the authors a chance to verify any possible misconceptions on my end:
    - If these are the embeddings used in the CLIP contrastive cosine distance loss, then it should be specified whether they are normalized inner products. The CLIP contrastive loss encodes distances only in the angular difference of the embeddings, so the embedding norms should not contribute to the analysis unless there's a good reason to.
    - If the used embeddings correspond to CLIP's normalized inner product space, then we also have to be careful about applying any sort of euclidean analysis, because the clustering from the learned representation probably lies on the surface of the N-dimensional normalized sphere.
- Is it possible to verify that there is no dataset leakage in the zero-shot analysis? Based on my understanding of the dataset collection pipeline, there should be existing captions from previous datasets such as MSCOCO and FLICKR30K included in this dataset, with possibly different image pairings. Can we confirm that none of the pre-trained models have been exposed to MSCOCO or FLICKR30k?

**Details Of Ethics Concerns:**

I'm not sure if this is a serious concern, but the free-use license for unsplash.com images does not cover the use of images with brands, works of art, or recognizable faces. The images in the paper certainly include examples of recognizable faces, which may be in breach of the license.

Update: Author responded and this is no longer a concern.

---

> ### Author Response · Authors · 2023-11-21
> **Response to Reviewer KVAJ (R4)**
>
> We thank reviewer $\textcolor{blue}{R4}$ for valuable comments on dataset collection process and the core challenge in this paper. We respond to the reviewer's concerns and questions in detail below.
>
> ## MPL2D and Maximize Diversity among Captions
>
> We thank reviewer $\textcolor{blue}{R4}$ for critical question on MPL2D and diversity maximization using sentenceBERT. We first apologize for the confusion we have made in the paper, and any potential misunderstanding of reviewer's question.
>
> > If the used embeddings correspond to CLIP's normalized inner product space
>
> We indeed used the normalized version for computing further statistics. We appreciate the concern when applying euclidean analysis with normalized inner product space. The aim of this metric is to measure how well each caption is paired with its image. We directly use the image enbedding as the pseudo center of its caption embeddings. This is because we argue that an image can have (potentially) infinitely many captions, which the collected captions are just a small subset of them.
>
> > whatever euclidean distance can be associated with in sentenceBERT's embedding representation
>
> To clarify the clustering process during caption selection we add further details about what is described in appendix A. We first extract the text embedding of all captions (including noisy captions) using sentenceBERT. We then proceed all the text embeddings to UMAP to learn a low-dimentional projection of the text embedding while maximally maintaining the structure of those embeddings in the original high-dimentional space. Noisy captions are also included to have more samples for better UMAP learning.
>
> After we obtain the UMAP projection, for each caption we select those captions which passed human-evaluation, and performing clustering in the low-dimentional 2D space. This is the commonly used visualization pipeline being used with different encoders and dimentionality reduction techniques. The results are also checked by human-annotators with randomly selected images. For image with more than 100 captions, we can see that each cluster represent different meanings. While for those with fewer captions (still larger than 10), the captions are still selected with maximal potential diversity.
>
> ## Contributions
>
> We thank reviewer $\textcolor{blue}{R4}$ for comments on our contributions.
>
> > I don't think the contributions of this work sufficiently shed light on any new aspects of the problem that the community was already aware of.
>
> To the best of out knowledge there are no works that address image polysemy in-depth. Our aim is to not only make the community aware of the issue, but to also make it possible to be able to quantify to what extent it is a problem. Our results show that polysemy has the potential to heavily degrade retrieval performance, which affects VLM applied to real-world data at large.
>
> ## Data Leakage
>
> We thank reviewer $\textcolor{blue}{R4}$ for valuable concern on the data leakage problem.
> We would like to clarify first that we did not include captions from MSCOCO or Flickr30k in IMP.
> We can only verify that image-caption pair in IMP does not exist in CC3M and CC12M.
> We observe that there exists handful cases of very similar captions (but not identical) existing in IMP and MSCOCO, but the image is not identical.
> We are not able to test data leakage of whether pre-trained models have been exposed to MSCOCO or Flickr30k, either due to the private dataset, or the massive public dataset which might need a lot of time to do the leakage test.

---

> > ### Comment · Reviewer_KVAJ · 2023-11-22
> > **Thanks for the response**
> >
> > Thank you for clarifying several details!
> >
> > I think the MPL2D metric, when computed on normalized vectors, is probably ok when the angular difference is small, because the squared euclidean distance is at least proportional to the cosine angle (I'm not sure if squared or unsquared euclidean is being used here). A more principled approach would have been to perhaps compute the intrinsic means and variances using the geodesics along the surface of the hypersphere [1,2]. Regardless, I suspect the current results would still hold, but I imagine the numbers would come out quite differently, and that the current approach probably wouldn't be accurate enough to assess fine-grained differences in such statistics.
> >
> > Regarding the use of sBERT, I agree that the overall formulation is reasonable and the additional quality control step suggests that the results are probably good enough.
> >
> > I would also like to thank the authors for checking the issue regarding data leakage. There's certainly a limited extent to which one can perform this test these days on large pretrained models, but it's still important to take note of when leakage might be an issue.
> >
> > Finally, as the authors indicated they intend to publicly release the data, I noticed that the Unsplash license (https://unsplash.com/terms) does not apply to:
> > - Trademarks, logos, or brands that appear in Photos
> >
> > - People’s images if they are recognizable in the Photos
> >
> > - Works of art or authorship that appear in Photos
> >
> > Have the authors accounted for this?
> >
> >
> >
> > [1] Oldfield et al. Parts of Speech-Grounded Subspaces in Vision-Language Models. Neurips2023
> > [2] Fletcher et al. Principal geodesic analysis for the study of nonlinear statistics of shape. IEEE Transactions on Medical Imaging ( Volume: 23, Issue: 8, August 2004)

---

> > > ### Author Response · Authors · 2023-11-22
> > > **Response to Reviewer KVAJ (R4)**
> > >
> > > We are grateful, and thank reviewer $\textcolor{blue}{R4}$, for the additional comments and suggestions. We are happy to provide further clarification:
> > >
> > > > I'm not sure if squared or unsquared euclidean is being used here
> > >
> > > For the calculation of MPL2D we have used squared euclidean distance.
> > >
> > > > A more principled approach - intrinsic means and variances
> > >
> > > We recognise the importance of using a more principled metric such as geodesic distance [1,2] as suggested by $\textcolor{blue}{R4}$. Since the vector is normalized, we focus on the geodesic distance on a hypersphere and compute the mean and variance. We report the result (CLIP-B/32) in table below:
> > >
> > > | Dataset | Geodesics mean| Geodesics var (scaled by $10^3$)|
> > > | -------- | -------- | -----
> > > | COCO     | 1.257       |2.189 |
> > > |FLickr30k | 1.247| 1.418 |
> > > |IMP-5k| 1.314 |2.837 |
> > >
> > > We will add this table to the paper and expand it to include the other datasets.
> > >
> > > > it's still important to take note of when leakage might be an issue
> > >
> > > We fully agree, and we are glad we could confirm that there is no data leakage between IMP and CC3M or CC12M, nor with MSCOCO or Flickr. If desirable we could explore this further on even larger datasets for the camera-ready version. However, given the uniqueness of our data pipeline we suspect that there will be limited to no leakage even with webscale datasets.
> > >
> > > > Have the authors accounted for [the Unsplash terms]?
> > >
> > > We appreciate the reviewer taking the time to look into the data and the source. While indeed the general Unsplash terms specify this, we are making use of the [Unsplash lite-dataset](https://unsplash.com/data) released for research purposes which has [specific terms](https://github.com/unsplash/datasets/blob/master/TERMS.md), that does not include this clause.
> > >
> > > Our understanding is that the lite dataset is available for both noncommercial and commercial use, as well as for the training of machine learning models. As such we believe this is a solid foundation to build upon, and we hope this makes it possible for the community at large to make use of IMP.

---

> > > > ### Comment · Reviewer_KVAJ · 2023-11-23
> > > >
> > > > Thank you for your very much prompt response on this, and sorry for being last-minute in my feedback. All immediate concerns have been addressed and I have no further questions.

---

### Official Review · Reviewer_FPXN · 2023-11-03

**Soundness:** 3 good
**Presentation:** 3 good
**Contribution:** 3 good
**Rating:** 8
**Confidence:** 4

**Summary:**

This paper presents the Image Polysemy dataset (IMP). IMP is designed to require models to understand that there are different plausible ways of describing an image that goes beyond the approach of crowdsourcing captions from the internet. IMP is collected in a multi-stage process starting from images sourced from Unsplash that includes human annotation and cleaning. The dataset is used to evaluate a broad collection of VLM in a zero-shot and fine-tuned setting. The results show that surprisingly good performance is possible in the zero-shot setting, and that fine-tuning various versions of the CLIP model can indeed improve performance. The experiments also include ablations on different CLIP-style models trained on different amounts of data, different architectures, and on different data sources.

Edit: thanks for your response to my questions!

**Strengths:**

1. Extensive zero-shot experiments across a large collection of models.
2. Should prove to be a useful resource for evaluating VLMs.

**Weaknesses:**

1. I didn't feel like both Figure 1 and Figure 3 needed to exist in the main body of the paper because it feels like they are duplicating information.
2. Not entirely clear why the dataset needed to be based on "high-quality stock photography from Unsplash".

**Questions:**

1. Why is the high-quality stock photography from Unsplash a good source for evaluating models in different scenarios?
2. Which steps did you take to ensure that none of the models had pretrained on the photos that you used from Unsplash?
3. Why did you use captions that described other images as the "human-authored" versions of the captions? Is CLIP similarity really good enough for this?
4. Why is RSUM a meaningful measure to report? It doesn't seem like it gives a better understanding of model performance than the original set-based retrieval measures.
5. What is the purpose of the qualitative analysis in Section 4.3? I couldn't fully understand the contribution of this section.

---

> ### Author Response · Authors · 2023-11-21
> **Response to Reviewer FPXN (R3)**
>
> We thank the reviewer $\textcolor{green}{R3}$ for insightful comments and recognition of IMP as a useful resource for evaluating VLM.
>
> ## Modification of the figure 1 and 3
>
> We thank the reviewer $\textcolor{green}{R3}$ for suggetion on better organizing the main text.
> We follow the reviewer's suggestion, now Figure 1 has been moved to appendix so there is no duplicated information.
>
> ## Choice of Unsplash Images
>
> We thank the reviewer $\textcolor{green}{R3}$ for insightful question on database we select.
> We would like to give two main reasons we chose Unsplash:
>
> 1.   These images from Unsplash, due to their nature (high resolution, less noise, aesthetic value), are more likely to be used as wallpapers or figures in blogs, forums, news, etc. This allows us to search for the use of this exact image over the internet and see how real humans use the image. On the other hand, images from other datasets are mostly likely to only appear once on the internet.
> 2.   We would like to make our dataset public, and Unsplash can be openly shared. So researchers can easily use the dataset for experiments or extend the dataset. For each image we have at least 10 perfect-matched image found by google-vision web entity search.
>
> ## Pre-training data, which contains Unsplash images
>
> We thank the reviewer $\textcolor{green}{R3}$ for valuable comment on potential data leakage.
> We can confirm that in CC3M and CC12M there is no image-text pair exists from IMP-5k. Since many models either pre-trained on private dataset or massive public dataset, we cannot test the data leakage for all these models. However, since our pipeline can get new image and captions from the web, the leakage problem should be at most a handful number.
>
>
> ## Human-authored captions
>
> We thank the reviewer $\textcolor{green}{R3}$ for critical comment on details of our caption gather process.
> While CLIP similarity is not sufficient to find semantically identical images fully, it did enable us to select candidate captions which we could then use within the further pipeline, including human annotation. Due to our pipeline we do finally end up with high quality captions. Our aim with this procedure was to increase the number of conceptual captions in our dataset, i.e., if a caption can be transferred from one image to a similar image then it is presumably less descriptive.
>
>
> ## Evaluation Metric
>
> We thank the reviewer $\textcolor{green}{R3}$ for valuable comment on evaluation metric.
> We are interested in learning how well the model can retrieve all its corresponding captions, which means the ability to deal with image polysemy. We largely agree with the reviewer that RSUM can not fully reflect this aspect. We have added medr and meanr for more analysis and more insights into the experiments, which is included in the shared response and in Appendix C.
>
>
> ## Qualitative analysis
>
> We thank the reviewer $\textcolor{green}{R3}$ for insightful question on qualitative analysis.
> In the qualitative analysis section, we have reported several “hard captions” and the false-negative rate from the image-text matching task. These hard captions passed the human evaluation, and the image-text pairs make sense to most annotators, but they failed to be matched by the model either zero-shot or after fine-tuning. Assuming we have a candidate image, there can be captions from other images which can be paired with the candidate image. However, since the candidate image and its captions has passed the human check, we would like to evaluate this by the false-negative rate, which should highlight the core difficulties for when models learn from polysemic images.

---

### Official Review · Reviewer_6sQn · 2023-11-05

**Soundness:** 3 good
**Presentation:** 3 good
**Contribution:** 3 good
**Rating:** 5
**Confidence:** 3

**Summary:**

This paper proposes IMP, a dataset for image-text pairs in which texts capture polysemy: diverse types of correspondences between each image and its (multiple) captions. The main difference from previous datasets is the curation and inclusion of multiple non-descriptive and naturally-occuring captions for a single image. The paper also reports results on this benchmark on image and text retrieval tasks.

**Strengths:**

- S1: I think that the research question studied in this paper is significant and interesting. This benchmark would be really beneficial to the community. As far as I can tell, no benchmark like this exists. Previous benchmarks also suffer from losing images over time.

- S2: Overall, both the approach for collecting the dataset and experimental design are sound.

**Weaknesses:**

- W1: Analysis of this dataset can be improved significantly from what is reported in Table 1. My major concern is in the characterization of polysemy, which is not much beyond “non-descriptive”. For instance, does any kind of image-text correspondence ontology emerge? Perhaps this needs human classification of (a subset) captions into a pre-defined categories, or automatically clustering captions, etc. This is especially important since quantitatively MPL2D cannot distinguish noise from diversity. Finally, I would also like to see further statistics beyond word lengths (word clouds, token/type ratio, etc. that would lead us to have a clearer picture of how this dataset is more diverse than existing ones.

- W2: The tasks that perform on this dataset are standard image- and text- retrieval tasks with zero-shot and fine-tuning evaluation. Further, an adaptation of models that address image polysemy does not lead to improved performance (see, e.g., SE models in Table 4). These experiments are a good start but the paper would be stronger with additional tasks that focus on measuring model’s capability in addressing polysemy such as image captioning generation given a “caption sense”.

- W3: Besides improving the analysis in W1 above, the discussion of image-text datasets can also be further expanded to include additional datasets in the analysis, including RedCaps and SBU captions.

- W4: Clarity on the data collection process is quite obscure (under Table 4). IMO, it is important to include the details on the database used to retrieve captions (CC3M and CC12M) in the main text so the reader is well-informed about the bias of these captions. In addition, it is important to include the details on how the authors optimize for diversity and quality control.

**Questions:**

At this point, what would change my mind the most is a much more thorough analysis of the dataset that focuses on polysemy (W1).

---

> ### Author Response · Authors · 2023-11-21
> **Response to Reviewer 6sQn (R2) (1/2)**
>
> We thank the reviewer $\textcolor{orange}{R2}$ for insightful comments and recognition of the benefit of this dataset to the community. In addition to what listed in the common response, we would like to response:
>
> ## Characterisation of polysemy
>
> We purposefully do not try to strictly define polysemy as any such definition would quickly exclude valid examples. In essence, image polysemy emerges when considering images in real-world scenarios: each person who observes an image may have a different interpretation, the existence of these different interpretations (or meanings) is what defines image polysemy. Polysemy does not necessarily have to be non-descriptive - however, we found that existing captions for images are mainly descriptive and by adding non-descriptive captions we were able to increase the diversity of our dataset. We do not that descriptive versus non-descriptive is not a binary, but rather a spectrum, and we did not find that any natural ontology emerges from the captions.
>
> Distinguishing between noise from diversity is indeed an open challenge, and MPL2D cannot do this in a straightforward manner indeed. However, both MSCOCO and IMP/5K consist of human annotated or human verified captions, which significantly lowers the noise present in these datasets. For a comparison between MSCOCO and IMP/5K we can thus conclude that differences in MPL2D are due to differences in diversity.
>
> For a large-scale webscraped dataset like Conceptual Captions the MPL2D score is most likely strongly influenced by noise - yet it gives us an understanding of the range of MPL2D and what a possible upper-bound may be.
>
> ## Experiments and future work
>
> We thank the reviewer $\textcolor{orange}{R2}$ for critical comments on the cross-modal retrieval task, evaluation, and suggestions.
>
> The intuition of having these experiments is given in the shared reply. We would like to answer further regarding:
>
> >   Further, an adaptation of models that address image polysemy does not lead to improved performance (see, e.g., SE models in Table 4).
>
> The SetEmbedding (SE) model uses slot attention to have different slots focusing on different regions of the given input. According to the SetEmbedding paper, it can be treated more as a multi-view approach and can achieve SOTA performance on MSCOCO and Flickr30k. Due to the nature of object-centric learning, slot-attention modules would enable the model to link different combinations of objects in the image to different captions. However, we have performed small-scale experiments of how the SE model learns to utilise the slots. For example, when given four descriptive captions and one conceptual caption, it would learn to link the conceptual caption to the background noise (one slot always appears to be dedicated to 'noise'). Increasing the number of slots can somehow mitigate the problem, but we argue that an image can potentially have infinitely many different meanings. So this would need further modification on the SE model to deal with image polysemy, as well as moving beyond the multi-view focus of current SE models, which is out of the scope of this paper.
>
> >   These experiments are a good start but the paper would be stronger with additional tasks that focus on measuring model’s capability in addressing polysemy such as image captioning generation given a “caption sense”.
>
> We appreciate the reviewers recognition of the benefits our current experiments. The central question in our work is to show that current contrastive learning-based models fail to deal with the polysemy challenge. Whilst testing other types of models may give further insight into the polysemy challenge, it also adds complexity due to the difference in how these models are trained. Retrieval directly maps onto contrastive-learning and the alignment between images and text representations - and we believe that if the model cannot overcome this challenge in image-to-text retrieval then image captioning would suffer a similar fate. Additional "sense" prompts may be used during generation, however, as we cannot determine the set of possible prompts apriori this would leave the polysemy challenge unsolved still.
>
>
> ## Expand the discussion to additional datasets and Data Collection Details
>
> We thank the reviewer $\textcolor{orange}{R2}$ for valuable suggestions on including more datasets for comparison and more details of diversity and quality control.
> We have included SBUcaptions in the table, the analysis is based on the entire SBUcaptions (849k). And due to the nature of RedCaps, doing analysis on only one topic will be biased. Instead we select all topics of year 2020, and perform the analysis. We still need some time to get statistics and analysis for RedCaps, once we finished we will add them to the main text.

---

> > ### Author Response · Authors · 2023-11-21
> > **Response to Reviewer 6sQn (R2) (2/2)**
> >
> > We control the annotation quality by instructing the annotators to avoid “over-thinking”. Annotators will first see the caption and then the image. The judgement is based on their first and second impression (directly finding the link or after a few seconds of thinking) of whether the caption and image can be a good pair. We find that this instruction helps reduce far-fetched pairings, but leaves the possibility for creativity and highly polsemic pairings. After that, inter-annotator agreement need to be made as well as the review-feedback loop.
> >
> > For diversity quality control, we have annotators to check examples from IMP-5k compared to all candidate captions. And for noisyIMP, annotators need to check extreme outliers in the sentenceBERT embedding space (possibly meaningless sentence), and go though random image examples with captions selected by clustering.

---

### Official Review · Reviewer_9GYX · 2023-11-09

**Soundness:** 2 fair
**Presentation:** 2 fair
**Contribution:** 2 fair
**Rating:** 5
**Confidence:** 3

**Summary:**

The paper argues that current models are mostly trained on datasets where an image has a single caption and collects a dataset covering both descriptive and conceptual captions and each image could have multiple captions. Experiments show that current models struggle on the dataset even after fine-tuning.

**Strengths:**

Most prior work seeks to have cleaner and more descriptive captions; it is interesting to see efforts on including more conceptual captions.

The evaluation shows that current models struggle with retrieving more abstract and conceptual captions, which raises an interesting and challenging problem.

**Weaknesses:**

**A.** It seems that the “image polysemy” considered in this paper mainly means: the model should have the ability to match images to both “descriptive” and “conceptual” captions; previous datasets such as COCO contains mainly descriptive captions.

However, I am not fully convinced that the collected datasets have more “conceptual” captions than the web image-text data such as CC3M, since many of the captions in the dataset come from web data. The MPL2D also does not indicate that the collected dataset is more “conceptual”.

The only advantage of the dataset seems to be that it has multiple captions per image while CC3M/12M does not. But if the final goal is to teach a model to match an image to descriptive / conceptual captions, then it is not clear why it is necessary to have multiple captions per image for training; as long as CC3M has a lot of conceptual & descriptive captions, then the model can learn to retrieve both types captions.
E.g., say there is a dataset A with 1K images each with 5 captions; suppose a dataset B with 5K images each with 1 caption. The captions in A and B are identical. Then I do not see the necessity of training on A if we have dataset B.

In sum, it would be better if the paper could illustrate either a) why / how the dataset has more diverse / conceptual captions than CC3M or b) the importance of having multiple captions per image.

**B.** The experiments are not very insightful. While the problem of image polysemy is interesting, the paper simply evaluates/fine-tunes current models with image-text retrieval on the collected datasets. The take-away conclusion seems to be that the task is hard and larger models perform better.
I would expect more analysis and discussions on why and how studying image polysemy could benefit future vision-language models and how to model such a phenomena. For example, does explicitly modeling image polysemy benefit other tasks that could require high-level conceptual understanding (e.g., understanding actions, events, memes, etc)?

**C.** For the image-to-text retrieval evaluation, how does this test model’s ability to handle image polysemy? If I understand it correctly, as each image has 5 matching captions, as long as the model retrieves 1 of the matching captions, then it is counted as correctly retrieved? Then the evaluation protocol does not test whether the model handles polysemy; if for most images, at least 1 of the captions is “descriptive”, then a model that only “understands” descriptive captions will still score high on image-to-text retrieval.

**D.** For comparing the dataset with CC3M/CC12M on MPL2D, why not treat the collected dataset as a single caption dataset (either downsample to one caption per image or just “duplicate” the images)?

In addition, I am not sure about the takeaway message by comparing MPL2D: is higher MPL2D score better or lower score better? On the one hand, if there are more diverse / conceptual captions, the score is higher; on the other hand, if the captions are noisier, the score is also higher. Thus, a lower/higher score could be attributed to these two possible factors and we cannot make a conclusion.

**Questions:**

Please see Weaknesses.

---

> ### Author Response · Authors · 2023-11-21
> **Response to Reviewer 9GYX (R1) (1/2)**
>
> We thank reviewer $\textcolor{red}{R1}$ for highlighting the importance of conceptual captions and their distinction from descriptive captions in our dataset. The reviewer's points raise crucial aspects of our work, which we aim to clarify further in addition to the shared response:
>
> ## Definition of "Image Polysemy":
>
> We thank reviewer $\textcolor{red}{R1}$ for valuable comments on "Image Polysemy".
>
> >   It seems that the “image polysemy” considered in this paper mainly means: the model should have the ability to match images to both “descriptive” and “conceptual” captions
>
> It’s correct that in our dataset, we can say captions are "descriptive" or "conceptual". A more structured way to classify captions is possible, and we would like to leave this to be discovered by models instead of human design.
>
> The concept of image polysemy in our work revolves around the inherent capacity of an image to convey multiple, diverse meanings. This multifaceted nature of images presents a significant challenge for current vision-language models, particularly those based on contrastive learning.
>
> As noted in our shared reply, the central aim of this paper is not to develop or expect models to match both descriptive and conceptual captions accurately. Instead, we focus on demonstrating that contrastive learning-based models struggle with the presence of multiple meanings expressed across captions.
>
> >   However, I am not fully convinced that the collected datasets have more “conceptual” captions than the web image-text data such as CC3M.
> >
> >   The MPL2D also does not indicate that the collected dataset is more “conceptual”.
>
> We would like to clarify that we are not expecting more “conceptual” captions than CC3M. We would expect a mixture of captions with different conceptual levels. The use of MPL2D is to illustrate the level of diversity per dataset instead of more “conceptual”. Moreover, we emphasise that our 5K test set has gone through human annotation, whereas datasets like CC3M have not. Any diversity in our IMP/5K thus comes from polysemy, whereas in CC3M it may still be noise.
>
> ## Importance of multiple captions per image:
>
> We thank reviewer $\textcolor{red}{R1}$ for insightful questions on the structure of the dataset.
>
> >   b) the importance of having multiple captions per image
>
> We believe that multiple captions differs from having a larger dataset with single captions per image. We show in the paper that when a model is exposed to multiple interpretations of the exact same image, it fails to recognise the multifaceted nature of visual content. The model could learn to match only one caption to the image if we use a single caption dataset like CC3M. To get a deep understanding of the challenge of image polysemy we believe it is necessary to explicitly model it in the dataset. We see this is a crucial step in advancing vision-language models beyond their current limitations, and beyond the typical paradigm of associating a single image with a single meaning.
>
> Training models on datasets with single captions per image, even if they include both descriptive and conceptual captions, does not provide the same learning opportunity. It does not explicitly teach the model that multiple equally valid captions can describe a single image, an essential aspect of understanding image polysemy.
>
> Moreover, creating a dataset with multiple captions per image allows us to control the experimental variables, thereby bridging datasets like CC3M and MSCOCO. By choosing a multiple-caption style, we facilitate a more balanced and reasonable comparison of how well models can handle diverse challenges not adequately addressed by existing datasets.
>
> In conclusion, the multiple captions per image in our dataset are not merely about increasing the quantity of data but are a strategic decision to advance the study of image polysemy. This approach provides a unique opportunity to understand how vision-language models handle real-world challenges such as image polysemy.

---

> > ### Author Response · Authors · 2023-11-21
> > **Response to Reviewer 9GYX (R1) (2/2)**
> >
> > ## Real-World Applicability of Vision-Language Models:
> >
> > We thank reviewer $\textcolor{red}{R1}$ for this insightful question regarding the rationale behind our focus on the retrieval task and the broader implications of studying image polysemy for future vision-language models. We touch upon the former point in the shared response, and expand on the latter in the following.
> >
> > >   I would expect more analysis and discussions on why and how studying image polysemy could benefit future vision-language models and how to model such a phenomena.
> >
> > 1.  **Increasing Prevalence of Polysemy in Real-World Data:**
> >     -   As we move towards utilising more real-world data in AI applications – such as in understanding events, memes, news, and social media content – the presence and significance of polysemy become increasingly prominent. Unlike controlled datasets like MSCOCO, real-world data is not bound by predefined usage constraints, leading to a natural occurrence of polysemy.
> >     -   Recognising and accurately interpreting this polysemy is critical for effectively deploying vision-language models in these real-world scenarios.
> > 2.  **Need for Models to Navigate Polysemy:**
> >     -   Our research underscores the necessity for vision-language models to handle image polysemy, especially in user-facing applications. While it might be feasible to focus on either descriptive or conceptual interpretations, our findings show that current models often fail to distinguish between these types effectively. This lack of distinction results in a degradation of model performance when faced with polysemous images.
> >     -   Addressing this challenge is crucial for advancing vision-language learning, ensuring that models are equipped to deal with real-world data's diverse and complex nature.
> > 3.  **Highlighting a Key Challenge in Vision-Language Learning:**
> >     -   We hope that by demonstrating the limitations of current models in handling image polysemy, our work brings to light a significant issue that has been somewhat overlooked in the field. This challenge is not merely academic but has profound implications for the practical application of AI in diverse and dynamic real-world contexts.
> >     -   We would like to see our findings serve as a call to action for the research community, emphasising the need to develop more sophisticated models capable of understanding and interpreting the nuanced meanings of images as they are used in everyday life.
> >
> >
> >
> > ## Evaluation methodology
> >
> > We thank the reviewer $\textcolor{red}{R1}$ for raising this critical point about our evaluation methodology. We have added more evaluation using MeanR and MedR in the paper in Appendix C, as mentioned in the shared reply. We agree that if the model finds the descriptive one from five captions, that would be counted, but if this were fully the case, then the recall performance should be much higher than the results shown in Tables 2, 3, and 4. As such the mere presence of diverse captions makes the task more challenging.
> >
> >
> > ## Comparing to CC3M/CC12M on MPL2D:
> >
> > We thank the reviewer $\textcolor{red}{R1}$ for valuable suggestions on comparing IMP with CC3M on MPL2D. We would like to further add this way of computation to table 1 for fair comparison.
> >
> >
> > | Dataset   | MPL2D mean | MPL2D std |
> > | --------- | ---------- | --------- |
> > | COCO      | 1.175      | 0.038     |
> > | FLickr30k | 1.168      | 0.031     |
> > | IMP-5k    | 1.221      | 0.042     |
> >
> >
> > By comparing the MPL2D scores of IMP and CC3M, we aim to understand how pre-trained models treat the representation of image-text pairs, and MPL2D offers a direct measure of this. A higher MPL2D score, in our context, suggests a richer, more nuanced representation of the image-text pair, aligning with our objective to challenge models with polysemous images.
> >
> > We also acknowledge that a high MPL2D score could potentially arise from noise within the dataset. For our human annotated test split IMP/5K this is not an issue, however, for other datasets such as CC3M this may indeed explain the higher score. Yet, the high score on this metric for the not-noisy IMP/5K does confirm the notion that this is a diverse set.
> >
> > Nonetheless, it remains challenging to distinguish noise from diversity, which is why we believe that approaches which try to reduce noise will ultimately not work, instead a more structural approach to image polysemy is needed.

---

### Author Response · Authors · 2023-11-21
**Common Response**

## Further details on the Role of Contrastive Learning in Image Polysemy and the Purpose of the Dataset

We are grateful for the insightful feedback provided by the reviewers. In this shared response we aim to expand on two critical aspects of our work: the role of contrastive learning in handling image polysemy and the intended purpose of our dataset, to answer common questions from the reviewers.

### Contrastive Learning and Its Limitations in Handling Polysemy ($\textcolor{red}{R1}$, $\textcolor{orange}{R2}$, $\textcolor{blue}{R4}$, $\textcolor{purple}{R5}$):

In many Vision-Language Models (VLMs), a prevailing assumption is that co-occurrence between images and text reflects a single, shared meaning. This assumption is particularly present in VLM trained with contrastive learning, where the training is optimised for aligning these two modalities. However, with its reliance on maximising the cosine similarity of two embedding vectors, this methodology inherently encounters difficulties in with the polysemic nature of images – a key challenge that our work seeks to explore.

Our research explores the extent to which VLMs, especially those trained with contrastive learning, are impacted by polysemy.  This limitation becomes particularly evident in our dataset, which challenges models to go beyond the typical one-to-one correspondence.
To directly evaluate VLMs trained in this manner we choose to focus on cross-modal retrieval tasks, as they give direct insight into the alignment between modalities.
Based on our findings we argue that the reliance on cosine similarity between individual vectors in retrieval tasks is a significant factor in why models underperform on our dataset.

By exploring these limitations, our work underscores the importance of re-evaluating how contrastive learning-based models are assessed, and VLM are trained, particularly in their ability to navigate the complexities of image polysemy. This evaluation is crucial for developing VLMs that are applicable and effective in real-world scenarios.

### Nature and Purpose of Our Dataset($\textcolor{red}{R1}$, $\textcolor{orange}{R2}$, $\textcolor{green}{R3}$, $\textcolor{blue}{R4}$, $\textcolor{purple}{R5}$):

Given the subjectivity of pairing images and text we would argue that any dataset of image-text pairs runs into the issue of image polysemy. However, a crucial difference between IMP and other datasets is that we make this issue explicit. Instead, in MS-COCO style datasets the presence of polysemy was severely reduced through the narrow scope and focus of the annotation process. Yet, polysemy is omnipresent in large-scale webscraped datasets (e.g., Conceptual Captions), but this is left implicit by having only a single caption per image. In IMP we make the issue of polysemy explicit by having multiple captions per image where each caption is stems from real-world use and represents a different image sense.

This design of IMP allows is to serve as a challenging test bed for evaluating the capabilities of current models in handling image polysemy. This distinction between explicit and implicit presence of polysemy is vital, as it reduces the number of experimental variables that influence the results. We believe IMP aligns with the broader objective of enhancing model performance in real-world contexts, where images are inherently subject to multiple interpretations.

Within the paper we have insufficiently focused on IMP as a test bed by not distinguishing clearly enough between the test split and the training validation splits. While these latter two are provided, they are primarily meant for fine-tuning, and to adapt models to any potential domain shift. Our 5k test set has been human annotated and any image-text pairing can thus be considered to not be noise, and rather an example of image polysemy.

## Changes to the paper

As part of the rebuttal we have implemented the following changes in our paper:

* An expanded discussion on the inherent link between contrastive learning and retrieval tasks elucidates why this approach is particularly relevant to our study, and additional experiments on zero-shot and fine-tuning retrieval evaluated by medr and meanr, which we hope can give further insights in Appendix C.

| Model               | I2t MedR | I2T MeanR | T2I MedR | T2I MeanR |
| ------------------- | -------- | --------- | -------- | --------- |
| CLIP-L/14           | 22       | 1200      | 169      | 1301      |
| CLIP-L/14(finetune) | 11       | 175       | 18       | 669       |
| SE-101(finetune)    | 12       | 107       | 15       | 531       |
| SE/32(finetune)     | 11       | 112       | 14       | 527       |

* A redefinition of the dataset sections, differentiating the annotated IMP-5K (test) set from the noisyIMP-17K (train) and noisyIMP-1K (val) sets, to highlight the primary focus on the test component.
* Evaluate MPL2D on the entire pre-training dataset instead of a 125k subset.

---

> ### Author Response · Authors · 2023-11-21
> **Common Response (Continue)**
>
> We sincerely hope that these changes better reflect the intentions and implications of our work. We hope this paper can make the community aware of the importance of image polysemy and its impact on VLMs, as well as the limitations of the assumption of semantic equivalance between webscraped image-text pairs. We look forward to the opportunity to engage with the community in a broader dialogue.
>
> ---
>
> For brevity, we refer to reviewer $\textcolor{red}{9GYX}$ as $\textcolor{red}{R1}$, $\textcolor{orange}{6sQn}$ as $\textcolor{orange}{R2}$, $\textcolor{green}{FPXN}$ as $\textcolor{green}{R3}$, $\textcolor{blue}{KVAJ}$ as $\textcolor{blue}{R4}$, and $\textcolor{purple}{VqT9}$ as $\textcolor{purple}{R5}$.

---

### Meta-Review · Area_Chair_nZZY · 2023-12-12

**Metareview:**

This paper introduces a new dataset IMP, which is designed to challenge and evaluate vision-language models on image polysemy.

Reviewers agree this is an important topic to better understand how vision-language model represents the different concepts from the image. And the new dataset could benefit the community.

One major concern shared by multiple reviewers is that the task (image polysemy) is not well defined, thus leads to the question of how the new dataset represents the image polysemy. A more narrow but well defined case will be helpful to present.

Other comments from reviewers including:
1. More clarification on the dataset collection process.
2. More discussion and analysis of the dataset, including the categories of the captions, the difficulty of the task.
3. Suggestion or solution to improve the vision-language model on this polysemy images.

The authors actively participated the rebuttal discussion, but the major concern still remains.

In sum, the paper seems not in a state to be published yet. We encourage the authors to consider the feedbacks and resubmit it.

**Justification For Why Not Higher Score:**

See comments section.

**Justification For Why Not Lower Score:**

N/A

---

### Decision · Program_Chairs · 2024-01-16

Reject